



# Ammonium nitrate promotes sulfate formation through uptake kinetic regime

Yongchun Liu[1,5*], Zeming Feng[1], Feixue Zheng[1], Xiaolei Bao[2,7*], Pengfei Liu[3], Yanli Ge[3], Yan Zhao[3], Tao Jiang[4], Yunwen Liao[5], Yusheng Zhang[1], Xiaolong Fan[1], Chao Yan[6],

Biwu Chu[3,6], Yonghong Wang[6], Wei Du[6], Jing Cai[6], Federico Bianch[6], Tuukka Petäjä[6,8], Yujing Mu[3], Hong He[3] and Markku Kulmala[1,6]

1. Aerosol and Haze Laboratory, Advanced Innovation Center for Soft Matter Science and Engineering, Beijing University of Chemical Technology, Beijing, 100029, China

2. Hebei Provincial Academy of Environmental Sciences, Shijiazhuang, 050037, China

3. State Key Joint Laboratory of Environment Simulation and Pollution Control, Research Center for Eco-Environmental Sciences, Chinese Academy of Sciences, Beijing, 100085, China

4. Hebei Provincial Meteorological Technical Equipment Center, Shijiazhuang, 050021, China

5. College of Chemistry and Chemical Engineering, China West Normal University, Nanchong,

637002, China

6. Institute for Atmospheric and Earth System Research/Physics, Faculty of Science, University of Helsinki, P.O. Box 64, FI-00014, Finland

7. Hebei Chemical & Pharmaceutical College, Shijiazhuang, 050026, China

8. Joint International Research Laboratory of Atmospheric and Earth System Sciences

(JirLATEST), University of Helsinki and Nanjing University, Nanjing 210023, China

*Corresponding to:* Yongchun Liu (liuyc@buct.edu.cn) or Xiaolei Bao (bxl5@163.com)





## Abstract:

Although the anthropogenic emissions of $SO_2$ have decreased significantly in China, the decrease in $SO_4^{2-}$ in $PM_{2.5}$ is much smaller than that of $SO_2$. This implies an enhanced formation rate of $SO_4^{2-}$ in the ambient air, and the mechanism is still under debate. This work investigated the formation mechanism of particulate sulfate based on statistical analysis of long-term observations in Shijiazhuang and Beijing supported with flow tube experiments. Our main finding was that the SOR was exponentially correlated with ambient RH in Shijiazhuang (SOR=0.15+0.0032×exp(RH/16.2)) and Beijing (SOR=-0.045+0.12×exp(RH/37.8)). In Shijiazhuang, the SOR is linearly correlated with the ratio of aerosol water content (AWC) in $PM_{2.5}$ (SOR=0.15+0.40×AWC/$PM_{2.5}$). Kinetics studies suggest that uptake of $SO_2$ instead of oxidation of S(IV) in particle-phase is the rate determining step for sulfate formation. $NH_4NO_3$ plays an important role in the AWC and the transition of particle phase, which is a crucial factor determining the uptake kinetics of $SO_2$ and the enhanced SOR during haze days. Our results show that $NH_3$ significantly promoted the uptake of $SO_2$, subsequently, the SOR, while $NO_2$ had little influence on $SO_2$ uptake and SOR in the presence of $NH_3$.



## 1. Introduction

Atmospheric particulate matter (PM) is a world-wide concern due to its adverse effect

on human health, such as association with respiratory and cardiovascular diseases, lung

cancer and premature death (WHO, 2013;Lelieveld et al., 2015). The Chinese

government has made great efforts to improve the air quality (Cheng et al., 2019). For

example, the annual $PM_{2.5}$ concentration in Beijing decreased from 89.5 $\mu g\ m^{-3}$ in 2013

to 58 $\mu g\ m^{-3}$ in 2017 due to the stringent reduction of local and regional emissions

(Cheng et al., 2019;Ji et al., 2019). However, the $PM_{2.5}$ concentrations in most regions

of China (Cheng et al., 2019;Chen et al., 2019c;Huang et al., 2019;Tian et al., 2019) are

still significantly higher than the $PM_{2.5}$ standard recommended by World Health

Organization (WHO) (WHO, 2006). Haze events also occur with high frequency,

especially, in autumn and winter.

     Secondary aerosol, including sulfate ($SO_4^{2-}$), nitrate ($NO_3^-$), ammonium ($NH_4^+$)

(SNA) and secondary organic aerosol (SOA) usually contributes to ~70 % of $PM_{2.5}$

mass concentration in different regions (Huang et al., 2014;An et al., 2019). SNA often

accounts for more than a half of $PM_{2.5}$ mass in severe pollution events (Zheng et al.,

2015;Wang et al., 2016). Even $SO_4^{2-}$ exceeds more than 20 % of $PM_{2.5}$ mass (Guo et al.,

2014;Wang et al., 2016;Xie et al., 2015;He et al., 2018). Interestingly, the anthropogenic

emissions of $SO_2$ in 2017 reduced by ~90 % when compared with 2000 in Beijing

(Cheng et al., 2019;Lang et al., 2017). However, the decrease rate of particulate $SO_4^{2-}$

concentration (Lang et al., 2017;Li et al., 2017) is much smaller than $SO_2$ (Lang et al.,

2017;Zhang et al., 2020). This implies an enhanced oxidation rate of $SO_2$ in the



atmosphere (Lang et al., 2017). However, the mechanisms and kinetics of particulate $SO_4^{2-}$ formation in the real atmosphere are still open questions in many regions of China although they have been extensively discussed (Ervens, 2015;Warneck, 2018).

Particulate $SO_4^{2-}$ can be formed through homogeneous oxidation of $SO_2$ by
hydroxyl radicals (OH) and Stabilized Criegee Intermediates (SCIs) (Seinfeld and Pandis, 2006;Liu et al., 2019a). Modeling studies greatly underestimated (~54.2 %) $SO_4^{2-}$ concentration in serve pollution events in Beijing if only considering gas-phase oxidation of $SO_2$, while the normalized mean bias (NMB) decreased significantly after heterogeneous oxidation of $SO_2$ being considered (Zheng et al., 2015). Several
heterogeneous and/or multiphase oxidation pathways, such as oxidation of $SO_2$ or sulfite by $NO_2$ (He et al., 2014), $H_2O_2$ (Huang et al., 2015;Maaß et al., 1999;Liu et al., 2020a), HONO (Wang et al., 2020a) and $O_3$ (Maahs, 1983) or photochemical oxidation of $SO_2$ (Yu et al., 2017;Xie et al., 2015) on dust surfaces, catalytic oxidation of $SO_2$ by transition metal ions (TMI) (Warneck, 2018;Martin and Good, 1991;Wang et al., 2021)
and oxidation of $SO_2$ by $NO_2$ (Clifton et al., 1988;Wang et al., 2016;Cheng et al., 2016;Wu et al., 2019) in aqueous phase and heterogeneous oxidation of $SO_2$ on black carbon (Zhao et al., 2017;Zhang et al., 2020;Yao et al., 2020), have been proposed based on field measurements, laboratory and modeling studies. However, it is still controversial about the relative contribution of these pathways to $SO_4^{2-}$ production. For
example, the contribution of heterogeneous oxidation to $SO_4^{2-}$ production had been evaluated to be (48±5) % based on oxygen isotopic measurements (He et al., 2018), while it was 31 % even in the nighttime calculated by an observe based modeling (OBM)



simulation (Xue et al., 2016). Oxidation by OH could explain 33-36 % of $SO_4^{2-}$

production in BTH (Liu et al., 2019a), while it was negligible based on isotopic

measurements (He et al., 2018) and OBM simulation (Xue et al., 2016). As for the

oxidation of S(IV) species, which includes $SO_2$, $HSO_3^-$ and $SO_3^{2-}$, in aqueous phase,

oxidation by $H_2O_2$ (Liu et al., 2020b;Liu et al., 2020a), $NO_2$ (Wang et al., 2020a;Wang

et al., 2016;Cheng et al., 2016), $O_3$ (Fang et al., 2019), or TMI ($Mn^{2+}$) (Wang et al.,

2021) was proposed as the most important pathway by different researchers. It should

be noted that some reaction mechanisms mentioned above were proposed based on case

studies in short-term observations. Thus, long-term observations at different

environments are required to verify whether these mechanisms are statistically

important. In addition, the previous studies mainly focused on oxidation process of $SO_2$

in particle phase, while it is unclear what the control factors are from gas-phase $SO_2$ to

particle-phase sulfate. In particular, it has been found that the mass fraction of $NO_3^-$ and

$NH_4^+$ is increasing gradually (Lang et al., 2017;Li et al., 2018). It is still poorly

understood about the feedback between aerosol physics and aerosol chemistry.

In this work, one-year field observations have been performed in Shijiazhuang and

Beijing, synchronously. The formation mechanism of particulate sulfate has been

statistically discussed. The conversion ratio of $SO_2$ to sulfate is statistically and linearly

correlated to the aerosol water content (AWC), which is mainly modulated by

particulate ammonium nitrate. The reaction kinetics and other factors affecting sulfate

production have also been discussed.

## 2. Material and methods



**2.1 Field measurements.** Field measurements were performed at Shijiazhuang University (SJZ, Lat. 38.0281º and Lon. 114.6070º) and the west campus of Beijing University of Chemical Technology (BUCT, Lat. 39.9428º and Lon. 119.2966º) from March 15, 2018 to April 15, 2019. The SJZ station is on a rooftop of the main teaching building (5 floors, ~23 m above the surface), which is around 250 m from the Zhujiang road of Shijiazhuang. The BUCT station is on a rooftop of the main building (5 floors, ~18 m above the surface), which is around 550 m from the $3^{rd}$ ring road of Beijing. The distance between the two stations, which are the representative cities of BJH, is 260 km (Figure S1). Both stations are surrounded by traffic and residential emissions, thus, are typical urban observation sites. The details about the observation stations have been described in our previous work (Liu et al., 2020e;Liu et al., 2020d;Liu et al., 2020c).

Ambient air was drawn from the roof of the corresponding building. At the SJZ station, the mass concentration of $PM_{2.5}$ was measured by a beta attenuation mass monitor (BAM-1020, Met One Instruments, USA) with a smart heater (Model BX-830, Met One Instruments Inc., USA) to control the RH of the incoming air to 35% and a $PM_{2.5}$ inlet (URG) to cut off the particles with diameter larger than 2.5 μm. Particle phase Fe and Mn were measured using a heavy metal analyzer (EHM-X100, Skyray Instrument). Water-soluble ions ($Na^+$, $K^+$, $Mg^{2+}$, $Ca^{2+}$, $NH_4^+$, $SO_4^{2-}$, $Cl^-$ and $NO_3^-$) in $PM_{2.5}$ and gas pollutants (HCl, HONO, $HNO_3$, $SO_2$ and $NH_3$) were measured using an analyzer for Monitoring Aerosols and Gases (MARGA, ADI 2080, Applikon Analytical B.V., Netherlands) with 1 hour of time resolution. At the BUCT station, the mass concentration of $PM_{2.5}$ was the mean concentration obtained from four surrounding





monitoring stations (including Wanliu, Gucheng, Wanshouxigong and Guanyuan) of China Environmental Monitoring Centre (http://www.cnemc.cn). The chemical composition of $PM_{2.5}$ was measured using a Time-of-Flight Aerosol Chemical Speciation Monitor (ToF-ACSM, Aerodyne) after the ambient air went through a $PM_{2.5}$ inlet (URG) and a Nafion dryer (MD-700-24, Perma Pure). The configuration and the operation protocol of ToF-ACSM have been described well in previous work (Fröhlich et al., 2013). IE calibration for ACSM was performed using 300 nm dry $NH_4NO_3$ every month. Ambient air was drawn from the roof using a Teflon sampling tube (BMET-S, Beijing Saak-Mar Environmental Instrument Ltd.) with the residence time <10 s for gas-phase pollutant measurements. Trace gases including $NO_x$, $SO_2$, CO and $O_3$ were measured with the corresponding analyzer (Thermo Scientific, 42i, 43i, 48i and 49i) at both the SJZ and BUCT stations. Meteorological parameters including temperature, pressure, relative humidity (RH), wind speed and direction were measured using weather stations (WXT 520 at HAS/SJZ station and AWS 310 at AHL/BUCT station, Vaisala).

**2.2 Uptake kinetics of $SO_2$ on dust internally mixed with $NH_4NO_3$.** To understand the influence on RH on uptake kinetics of $SO_2$, the $\gamma_{SO2}$ on dust internally mixed with $NH_4NO_3$ was measured using a coated-wall flow tube reactor. The configuration of the reactor and data process protocol have been described in detail previously (Han et al., 2013;Liu et al., 2015). The $\gamma$, presenting the mass transfer kinetic of gas phase $SO_2$ to particle phase, is defined by the net loss rate of $SO_2$ per collision onto the surface (Ravishankara, 1997;Usher et al., 2003), namely,



$$\gamma_{obs} = \frac{\frac{-dc}{dt}}{\omega} = \frac{2k_{obs}r_{tube}}{<c>} \quad (1)$$

where $-dc/dt$ is the net loss rate of $SO_2$ when the surface is exposed to $SO_2$ (molecules

$s^{-1}$); $\omega$ is the collision frequency ($s^{-1}$); $k_{obs}$, $r_{tube}$ and $<c>$ are the first-order rate constant

of $SO_2$, the flow tube radius and the average molecular velocity of $SO_2$, respectively. A

correction for gas phase diffusion limitations was considered for $\gamma_{obs}$ calculations using

the Cooney–Kim–Davis (CKD) method (Cooney et al., 1974;Murphy and Fahey, 1987).

The BET uptake coefficients ($\gamma_{SO2,BET}$) was obtained from the mass dependence of $\gamma_{obs}$

as follows (Han et al., 2013;Liu et al., 2015):

$$\gamma_{SO2,BET} = [\text{slope}]\frac{A_g}{S_{BET}} \quad (2)$$

where [slope] is the slope of the plot of $\gamma_{obs}$ versus the sample mass in the linear regime

($mg^{-1}$); $A_g$ is the inner surface area of the sample tube ($cm^2$); and $S_{BET}$ is the specific

surface area of the particle sample ($cm^2\ mg^{-1}$).

Similar to a previous work (Zhang et al., 2019), dust internally mixed with

$NH_4NO_3$ was used in the kinetics study because it was difficult to deposit enough real

ambient particles onto the inner surface of the sample holder. Although the composition

of the model particles is much simpler than that of ambient particles, it is still

meaningful because we mainly focused on the influence of RH or aerosol water content

(AWC) on uptake kinetics of $SO_2$. The mixture (mass ratio = 2:1) of A1 Ultrafine test

dust (Powder Technology Inc.) and $NH_4NO_3$ (AR, Sinopharm Chemical Reagent Co.

Ltd, China) were suspended in the mixture of ethanol and water (v:v=1:3). The inner

surface of the Pyrex quartz tube (sample holder) was uniformly coated by the above

mixture and dried overnight in an oven at 393 K. The sample mass was calculated



according to the weighted mass of the dry tube before and after coating. To avoid the wall loss of $SO_2$ on the sample holder, all the inner surface of the sample holder was covered with particles. The wall loss of $SO_2$ on the remained surface (the inner surface of the outside tube and the outside surface of the sample holder) was subtracted in a

steady-state at the corresponding RH before the uptake experiment as done in our previous work (Liu et al., 2015). The mean concentrations of $SO_2$, $NO_2$ and $NH_3$ were 8.3±5.2 (0.4-49.1), 31.5±13.2 (2.5-85.1) and 41.0±18.4 (0.3-126.4) ppb, respectively, in polluted events (with the $PM_{2.5}$ concentration higher than 75 $\mu g\ m^{-3}$ and the RH less than 90%) in Shijiazhuang. The initial concentrations of $SO_2$, $NO_2$ and $NH_3$ in the

reactor were 190 ± 2.5, 100 ± 2.5 and 50 ± 2.5 ppb, respectively. The initial concentrations of $NO_2$ and $NH_3$ were close to their ambient concentrations, while a high initial $SO_2$ concentration was used here to obtain a good signal to noise ratio for $\gamma_{SO2}$ measurements. In this work, we aimed to understanding the influence of AWC on the uptake kinetics of $SO_2$. Therefore, we fixed the initial concentrations of pollutants and

the temperature at 300 K. $SO_2$ and $NO_2$ were measured using the corresponding analyzer (Thermo 43i and 42i) and $NH_3$ was measured by an ammonia analyzer (EAA-22, LGR, USA). The specific surface area of the mixture of A1 dust and $NH_4NO_3$ was 0.813 $m^2 \cdot g^{-1}$, measured by a nitrogen Brunauer-Emmett-Teller (BET) physisorption analyzer (Quantachrome Autosorb-1-C). RH from 0 to 80 % was adjusted by varying

the ratio of dry to wet zero air (water bubbler) and measured by a RH sensor (HMP110, Humicap). Control experiments demonstrate that adsorption of $SO_2$ on the quartz tube is negligible.





**2.3 Calculations of AWC, aerosol pH and production rates of sulfate in aerosol liquid water.** The AWC and aerosol pH in Shijiazhuang were calculated using the

ISORROPIA II model using the measured concentrations of $SO_4^{2-}$, $NH_4^+$, $NH_3$, $NO_3^-$,

HNO$_3$, Cl$^-$, HCl, Na$^+$, Ca$^{2+}$, K$^+$ and Mg$^{2+}$, RH and temperature as input. The particles

were assumed in metastable phase using a forward method (Song and Osada, 2020;Shi

et al., 2019). The dataset with RH lower than 35 % were excluded (Pye et al., 2020) due

to large uncertainties of aerosol pH (Ding et al., 2019;Guo et al., 2016;Pye et al., 2020).

pH was then calculated according to:

$$pH = -\log_{10} \frac{1000\gamma_{H^+}c_{H^+}}{AWC} \quad (1)$$

where $\gamma_{H^+}$ is the activity coefficient of H$^+$. The AWC of model particles for laboratory

studies was also calculated with the known composition, while the aerosol pH in Beijing

were not calculated because the concentrations of Na$^+$, Ca$^{2+}$, K$^+$ and Mg$^{2+}$ were

unavailable.

Similar to previous studies (Liu et al., 2020a;Cheng et al., 2016), four oxidation

pathways of S(IV) in aqueous-phase were accounted for, i.e., oxidation by O$_3$, H$_2$O$_2$,

NO$_2$ and TMI (Fe$^{3+}$ and Mn$^{2+}$), according to following equations:

$$-\left(\frac{d[S(IV)]}{dt}\right)_{O_3} = \left(k_0[SO_{2,aq}] + k_1[HSO_3^-] + k_2[SO_3^{2-}]\right)[O_{3,aq}] \quad (3)$$


$$-\left(\frac{d[S(IV)]}{dt}\right)_{H_2O_2} = \frac{k_3[H^+][HSO_3^-][H_2O_2,aq]}{1+K[H^+]} \quad (4)$$

$$-\left(\frac{d[S(IV)]}{dt}\right)_{TMI} = k_4[H^+]^\alpha[Mn^{2+}][Fe^{3+}][S(IV)] \quad (5)$$

$$-\left(\frac{d[S(IV)]}{dt}\right)_{NO_2} = k_5[NO_2,aq][S(IV)] \quad (6)$$

where $k_0 = 2.4 \times 10^4\,M^{-1}\,s^{-1}$, $k_1 = 3.7 \times 10^5\,M^{-1}\,s^{-1}$, $k_2 = 1.5 \times 10^9\,M^{-1}\,s^{-1}$, $k_3 = 7.45 \times 10^7\,M^{-1}$

$s^{-1}$, $K = 13\,M^{-1}$, $k_4 = 3.72 \times 10^7\,M^{-1}\,s^{-1}$, and $\alpha = -0.74$ (for pH≤4.2) or $k_4 = 2.51 \times 10^{13}\,M^{-1}$


$s^{-1}$, and $\alpha = 0.67$ (for pH>4.2) and $k_5 = (1.24–1.67) \times 10^7 \, M^{-1} \, s^{-1}$ (for $5.3 \leq pH \leq 8:7$; the

linear interpolated values were used for pH between 5.3 and 8.7) at 298K (Clifton et al.,

1988;Liu et al., 2020a). $[O_3, aq]$, $[H_2O_2, aq]$ and $[NO_2, aq]$ were calculated according

to the Henry's constants, which are $1.1 \times 10^{-2}$, $1.0 \times 10^5$ and $1.0 \times 10^{-2} \, M \, atm^{-1}$ at 298 K

for $O_3$, $H_2O_2$ and $NO_2$, respectively. $H_2O_2$ concentrations were unavailable during our

observations. It was fitted based on temperature like a previous work (Fang et al., 2019)

and varied from 0.05 to 3.7 ppbv, with a mean value of 0.62±0.52 ppbv. The

concentrations of $Fe^{3+}$ and $Mn^{2+}$ were calculated according to the measured total Fe and

Mn concentrations assuming 18% of total Fe and 30 % of total Mn were soluble (Wang

et al., 2014;Cui et al., 2008) and the precipitation equilibriums of $Fe(OH)_3$ and

$Mn(OH)_2$ depending on pH. The concentrations of Fe and Mn before December 2018

were estimated according to their mean ratios to $PM_{2.5}$ mass concentration (Wang et al.,

2014) because the instrument was unavailable.

## 3. Results and discussion

**3.1 Variation of sulfate in $PM_{2.5}$.** Figure 1A shows the hourly mean mass concentration

of $PM_{2.5}$ measured at SJZ and BUCT stations from March 15, 2018 to April 15, 2019.

The mass concentration of $PM_{2.5}$ in Shijiazhuang generally coincided well with that in

Beijing. This highlights the regional characteristic of air pollution in BJH. However,

Shijiazhuang usually showed significantly higher $PM_{2.5}$ concentration than that in

Beijing. The hourly mean $PM_{2.5}$ concentration varied in the range of 0 - 650 $\mu g \, m^{-3}$ with

an annual mean concentration of 86.4 ± 77.8 $\mu g \, m^{-3}$. The corresponding values in

Beijing were 1.5 - 556 and 55.0 ± 51 $\mu g \, m^{-3}$. Particularly, the wintertime mass



concentration of PM$_{2.5}$ in Shijiazhuang was as around 2.4 times as that in Beijing. This

is well consistent with previous results that Shijiazhuang is suffering from more serious

air pollution (Chen et al., 2019b) because of its larger population of heavy industries

and more intensive emissions than Beijing (Chen et al., 2019a).

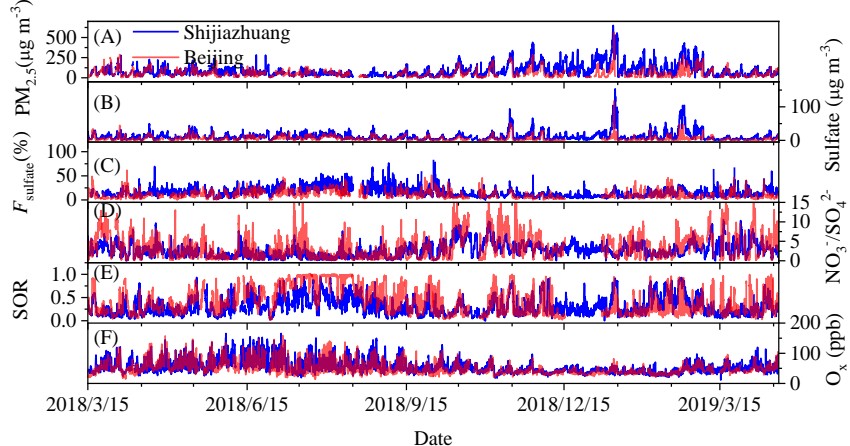

Fig. 1. The hourly mean (A) mass concentration of PM$_{2.5}$, (B) sulfate concentration, (C)

sulfate fraction in PM$_{2.5}$, (D) molar ratio of nitrate to sulfate, (E) sulfur oxidation ratio

(SOR) and (E) O$_x$ (=NO$_2$+O$_3$) concentration in Shijiazhuang and Beijing from March

15, 2018 to April 15, 2019.

Like the mass concentration of PM$_{2.5}$, both the mass concentration (Fig. 1B) and

the fraction of sulfate in PM$_{2.5}$ (Fig. 1C) in Shijiazhuang were usually higher than those

in Beijing. The annual mean sulfate concentrations in Shijiazhuang and Beijing were

11.7 ± 12.7 and 5.4 ± 6.9 μg m$^{-3}$, which annually contributed 15.3±8.7 % and 10.7±7.3 %

to the PM$_{2.5}$ mass concentrations, respectively. However, the molar ratio of NO$_3^-$ to

SO$_4^{2-}$ (3.37±3.05) corresponding to the mass ratio (2.17±1.97) in Beijing was

significantly higher than that in Shijiazhuang (2.69±1.80, corresponding to mass ratio



of 1.77±1.72) at 0.05 level. This is consistent with the emission inventories of air

pollutants, in which Shijiazhuang had larger $SO_2$ emissions than Beijing, and vice versa

for $NO_x$ emissions (Yang et al., 2019;Liu et al., 2017a;Chen et al., 2019a). A decrease

of sulfate concentration (5.4±6.9 μg m$^{-3}$) in Beijing was significant even when

compared with that in $PM_{1.0}$ (8.1±8.3 μg m$^{-3}$) measured from July 2011 to June 2012

(Sun et al., 2015), while the mass ratio of $NO_3^-/SO_4^{2-}$ (2.17±1.97) in Beijing showed an

obvious increase compared with those in 2011-2012 (1.3-1.8) (Sun et al., 2015) and

2008 (0.8-1.5) (Zhang et al., 2013). This can be ascribed to the effective reduction of

$SO_2$ emissions and the increased traffic emissions in Beijing.

The ground surface concentrations of pollutants are prone to be affected by

variation of mixing layer height (MLH) (Zhong et al., 2018;Tang et al., 2016). Sulfur

oxidation ratio (SOR), which is defined as the molar ratio of sulfate to total sulfur [41, 42],

$$SOR = \frac{n_{SO_4^{2-}}}{n_{SO_4^{2-}} + n_{SO_2}} \qquad (7)$$

was calculated and should be less affected by the MLH variation. As shown in Figure

1E, the SOR in Beijing was overall higher than that in Shijiazhuang. Thus, the annual

mean SOR in Beijing (0.42±0.29) was comparable with that reported in literatures

(Fang et al., 2019), while it was significantly higher than that in Shijiazhuang

(0.31±0.19) at 0.05 level. This implies the oxidation capacity in Beijing might be

stronger than that in Shijiazhuang or the air mass might be more aged in Beijing than

that in Shijiazhuang. However, the $O_x$ ($O_x = NO_2+O_3$) concentration in Shijiazhuang

was usually higher than that in Beijing (Fig. 1F). The annual mean $O_x$ concentration in

Shijiazhuang was 55.2 ± 22.3 ppb, which was significantly higher than that in Beijing





(50.7 ± 21.5 ppb) at 0.05 level. This means that a higher SOR should be observed in

Shijiazhuang than Beijing if gas phase oxidation mainly contributed to sulfate

formation. These results suggest that heterogeneous and/or multi-phase reactions play

important roles in particulate sulfate formation (Zheng et al., 2015;Martin and Good,

1991;Wu et al., 2019).

280       Figure 2A-C shows the mass concentration of $PM_{2.5}$ colored according to the mass

concentration of sulfate, the fraction of sulfate in the soluble PM and the SOR in

Shijiazhuang. In most serve pollution events, high $PM_{2.5}$ mass concentration well kept

pace with the high sulfate concentration, the fraction of sulfate and the SOR (colored

in grey color). For example, the mean $PM_{2.5}$ concentration was 411.7 ± 98.1 μg m$^{-3}$

during the pollution event occurred from 8:00 on January 12, 2019 to 0:00 on January

15, 2019. The corresponding sulfate concentration, fraction of sulfate in soluble PM

and SOR were 80.6 ± 24.0 μg m$^{-3}$, 39.4 ± 3.6 % and 0.79 ± 0.09, respectively. Other

pollution episodes, which were highlighted in grey color in Fig. 2, showed the similar

trend. The variations of the sulfate concentration, the fraction of sulfate in non-

refractory $PM_{2.5}$ and the SOR with $PM_{2.5}$ mass concentration in Beijing were similar to

Shijiazhuang and shown in Fig. S2. These results confirm that the conversion rate of

$SO_2$ to sulfate is promoted in pollution days when compared with that in clean days.

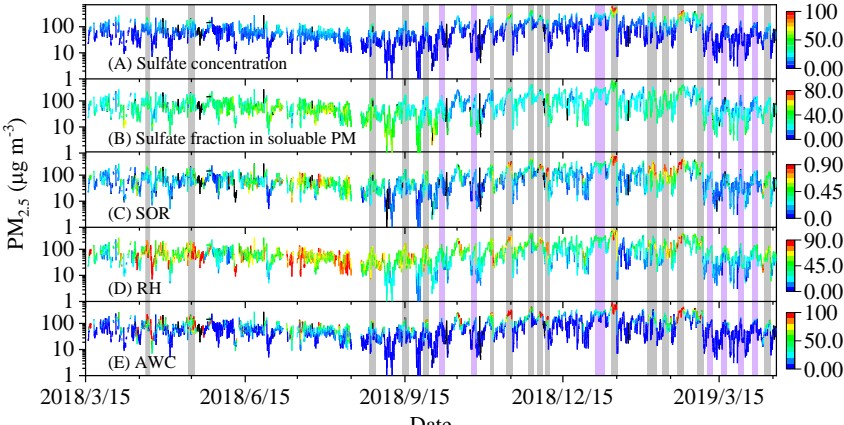

Fig. 2. Mass concentration of $PM_{2.5}$ colored according to (A) sulfate concentration, (B) sulfate fraction in soluble PM, (C) SOR, (D) RH and (E) AWC in Shijiazhuang. The shade areas in grey indicate the pollution events with high concentration of sulfate at high RH, while the purple ones the mean pollution events with low sulfate fraction at high RH.

**3.2 Role of aerosol water content in sulfate formation.** Previous studies have found that severe pollution events are frequently accompanied with high RH (Zhang et al., 2018;Tang et al., 2016;Wu et al., 2018;Liu et al., 2019b;Clifton et al., 1988;Maahs, 1983;Martin and Good, 1991). As shown in Fig. 2D, the high concentration of sulfate positively correlated with high RH in most cases, which were shaded in grey columns. However, some pollution events (shaded in purple columns) also occurred under high RH but the sulfate concentration or sulfate fraction in soluble PM was not so high. This means that high RH is a necessary but not a sufficient condition for sulfate conversion in severe haze pollution events. Thus, it is difficult to fully understand the general regularity behind the dataset or overemphasize the importance of a specific process in





the atmosphere based on case studies. This might be the reason why contrary

conclusions about the formation path of sulfate were drawn by different researchers.

We statistically analyzed the relationship between the SOR and the RH. All the hourly

mean data of the SOR and RH have been binned into 100×100 boxes. Then, the density

of data points, which statistically indicates the occurrence of the events at given values

of RH and SOR, was calculated using a bivariate kernel density estimator (Wand and

Jones, 1993).

Figure 3A and B show the 2D Kernel density graphs between the SOR and the RH

in Shijiazhuang and Beijing. The color bar shows the density of data points. Although

the SOR varied obviously at a certain RH, the most probable distribution of SOR could

be exponentially fitted as a function of RH in Shijiazhuang (Fig. 3A), that's,

SOR=0.15+0.0032×exp(RH/16.2) (R=0.79). This is consistent with the dependence of

SOR on RH based on previous studies (Tian et al., 2019;Wu et al., 2019). It should be

noted that both SOR and RH showed obvious diurnal variation (Fig. S3). Their diurnal

variations were somewhat similar, but a four-hours of time lag was observed between

their minimum values. This means that the diurnal variations of SOR and RH might

also contribute to the strong dependency of SOR on RH (Fig. 3A and B). However, the

exponential dependency of SOR on RH was still observable in the night or in the day

(Fig. S4A and B). It did so in winter or summer (Fig. S4C and D). This means that

aqueous reactions is important for sulfate formation even if the influence of diurnal and

seasonal variations are ruled out (Wang et al., 2016;Cheng et al., 2016).



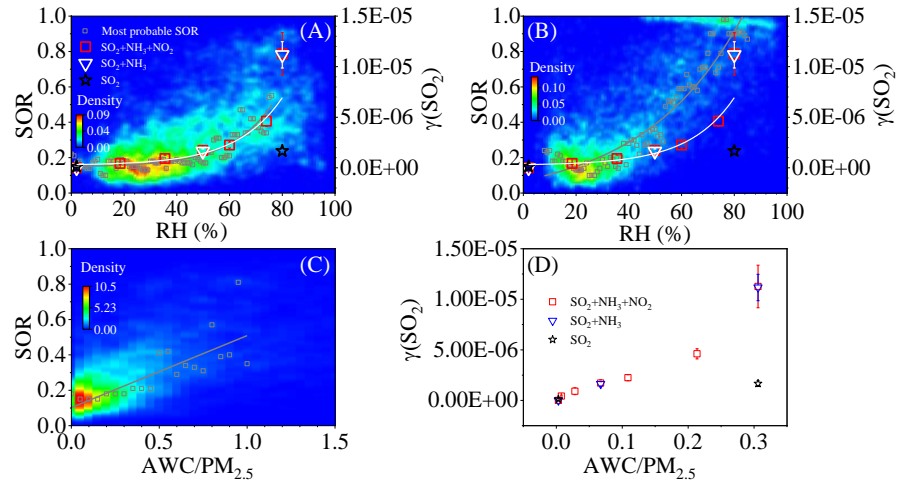


Fig. 3. Relationship between SOR and $\gamma_{SO2,BET}$ on dust internally mixed with $NH_4NO_3$

(2:1) and RH in (A) Shijiazhuang and (B) Beijing, and the correlation of (C) SOR in

Shijiazhuang and (D) $\gamma_{SO2,BET}$ with $AWC/PM_{2.5}$. The initial concentrations of $SO_2$, $NO_2$

and/or $NH_3$ in the flow tube reactor were $190 \pm 2.5$, $100 \pm 2.5$ and/or $50 \pm 2.5$ ppb,

respectively. The grey lines are the fitting curves for the most probable SOR and the

white lines are the fitting curves for the $\gamma_{SO2,BET}$.

In Fig. 3A, 72.5 % of the data points of Shijiazhuang (6509 over 8980 effective

points, which shown in small grey dots) were in the domain with the RH range of 10 %

- 70 % and the SOR range of 0.05 – 0.42, while 10.1 % of data points were in the region

with the RH greater than 70 % and the SOR greater than 0.42. The first region

corresponded to a lower mean $PM_{2.5}$ concentration, sulfate concentration and SOR

($76.1 \pm 62.78$ μg m$^{-3}$, of $8.1 \pm 6.3$ μg m$^{-3}$, and $0.21 \pm 0.09$, respectively) compared with the

second one ($115.7 \pm 96.7$ μg m$^{-3}$, $22.4 \pm 20.4$ μg m$^{-3}$ and $0.62 \pm 0.14$, respectively). As

shown in Fig. 3B, the SOR also exponentially increased as a function of RH in Beijing.

74.6 % of 8169 data points were in the first region. The mean $PM_{2.5}$ concentration,





sulfate concentration and SOR were $48.2 \pm 44.8$ µg m$^{-3}$, $2.9 \pm 3.0$ µg m$^{-3}$ and $0.21 \pm$ $0.10$ in the low RH region, while they were $69.9 \pm 50.9$ µg m$^{-3}$, $9.4 \pm 8.5$ µg m$^{-3}$ and $0.83 \pm 0.15$ in the high RH region. The most probable distribution of SOR in Beijing could also be exponentially fitted as a function of RH (SOR=-

$0.045+0.12 \times \exp(\text{RH}/37.8)$, R=0.92). However, the SOR was more sensitive to RH in Beijing than that in Shijiazhuang. This might be explained by the increased importance of sulfate formation via gas phase reactions in Beijing (Fang et al., 2019;Hollaway et al., 2019) because the PM$_{2.5}$ mass concentrations in Beijing were significantly lower than that in Shijiazhuang (Fig. 1).

Formation of particle phase sulfate through heterogeneous or multiple phase oxidations includes the uptake of SO$_2$ and the following oxidation in particle phase. Thus, it is meaningful to identify the rate determining step (RDS) for understanding the evolution of the SOR. As shown in Fig. 3, the initial $\gamma_{\text{SO2, BET}}$ increased exponentially from 0 to $(1.13 \pm 0.21) \times 10^{-5}$ when the RH increases from 2 % to 80 % in the presence

of $50 \pm 2.5$ ppb NH$_3$ with or without $100 \pm 2.5$ ppb NO$_2$. The dependence of $\gamma_{\text{SO2, BET}}$ on RH was $\gamma_{\text{SO2, BET}} = 2.44\text{E-}7 + 6.69\text{E-}8 \times \exp(\text{RH}/17.4)$ with a correlation coefficient of 0.96. A transition region of the $\gamma_{\text{SO2, BET}}$ verse the RH was observable when the RH ranged from 60 % to 80 %. When the RH was higher than 70%, the $\gamma_{\text{SO2, BET}}$ increased quickly as a function of the RH. The similar dependency on RH for the $\gamma_{\text{SO2, BET}}$ and the

SOR suggests that the uptake kinetic of SO$_2$ might determine sulfate formation.

In a previous work (Zhang et al., 2019), it has been found that all the uptake of SO$_2$ on dust or nitrate coated dust can be transformed into sulfate over the time scale of





uptake experiment using the similar coated-wall flow tube reactor. Another study also observed the quick formation of sulfate on the surface of aqueous microdroplets without the addition of other oxidants, which was explained by the direct interfacial electron transfer from $SO_2$ to $O_2$ on the aqueous microdroplets (Hung et al., 2018). This means that oxidation of S(IV) might not be a RDS of sulfate formation. The oxidation processes can be ascribed to catalytic oxidation by $O_2$ in the presence of transition metal, oxidation by $O_2$ and nitric acid promoted by protons in the presence of nitrate (Zhang et al., 2019), and the oxidation by other dissolved oxidants in liquid phase (Chen et al., 2019d;Cheng et al., 2016;Wang et al., 2016). To further validate this assumption, the formation rates of $SO_4^{2-}$ ($d[SO_4^{2-}]/dt$) in aerosol liquid phase were calculated according to the method used in previous work (Liu et al., 2020a;Cheng et al., 2016). If oxidation of S(IV) is the rate determining step, the formation rate should show a similar dependence on RH like the SOR.

As shown in Fig. S5, the relative contributions of different oxidation paths of S(IV) varied obviously case by case. In summer and autumn, oxidation by $H_2O_2$ was the most important path followed by TMI. In winter, however, either $NO_2$, $O_3$ or $H_2O_2$ could contribute to the major oxidation path. Figure 4A and B show the dependence of the formation rate of sulfate on RH in the range of 35%-100% in Shijiazhuang. The dataset for RH below 35 % were omitted due to the large uncertainty in aerosol pH calculations (Ding et al., 2019;Guo et al., 2016;Pye et al., 2020). The relative contributions of different oxidation paths of S(IV) also varied obviously as a function of RH. $NO_2$ and $O_3$ played important role in aqueous S(IV) oxidation when RH was from 35 % to 45%,





while TMI became the dominator when RH ranged from 45% to 70%. Above 70% RH,

the contribution of $H_2O_2$ was dominant, which is consistent with several recent studies

(Liu et al., 2020a;Liu et al., 2020b). However, the total formation rate of sulfate in

aerosol liquid phase slightly decreased as RH increasing. A weak downward trend of

the $d[SO_4^{2-}]/dt$ with RH was also observable in the 2D Kernel density graphs as shown

in Fig. 4B. This is opposite to the dependencies of the SOR and the $\gamma_{SO2}$ on RH as

discussed above, which means the RDS for sulfate formation should be the uptake of

$SO_2$ instead of oxidation of S(IV) in aqueous phase.

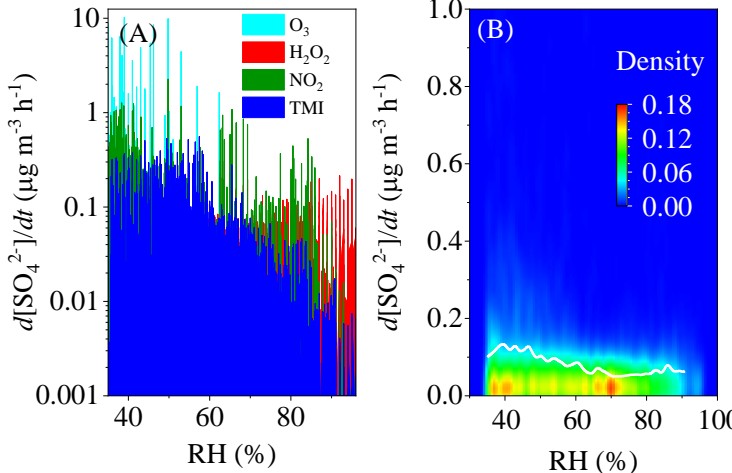

Fig. 4. Dependence of sulfate formation rates on RH in Shijiazhuang. The white line is

the probability weighted $d[SO_4^{2-}]/dt$.

Phase state is a crucial factor determining the mass transfer of pollutants from gas

phase to particle phase (Davis et al., 2015;Marshall et al., 2018;Shiraiwa et al., 2011;Liu

et al., 2014), while the AWC or RH greatly affects the phase state of aerosol particles

(Duan et al., 2019;Liu et al., 2019b;Shiraiwa et al., 2017). For example, ambient

particles were found to change from semi-solid to liquid state when the RH was above



~60 % (Liu et al., 2019b;Liu et al., 2017b) corresponding to the AWC higher than ~15

$\mu g\ m^{-3}$ (Liu et al., 2017b) under the typical urban environment in Beijing based on

rebound fractions measurements. It was also confirmed that haze particles displayed a

solid-aqueous equilibrium state when the RH was around 60-80% using an individual

particle hygroscopicity system (Sun et al., 2018). As shown in Fig. S6, the most

probable distribution of the AWC exponentially increased with the RH (AWC= -5.76

+ 5.15×exp(RH/36.1), R=0.98) in Shijiazhuang. An obvious transition region of the RH

between 60 % and 80 % was also observed. These results indicate that the liquid phase

aerosol should appear when the RH is higher than ~60 % (Liu et al., 2019b;Liu et al.,

2017b), subsequently, promote the conversion of $SO_2$ to sulfate. The SOR increased as

a power function of AWC (SOR = $0.072+0.043×AWC^{0.53}$, R=0.78), while it was

linearly correlated with the ratio of $AWC/PM_{2.5}$ (SOR = 0.15 + 0.40 × $AWC/PM_{2.5}$,

R=0.78) as shown in Fig. 3C. Similarly, the AWC of dust internally mixed with

$NH_4NO_3$ was also calculated using the ISORROPIA II model. The $\gamma_{SO2,BET}$ also showed

a similar trend as a function of $AWC/PM_{2.5}$ ($\gamma_{SO2,BET}$ = 3.08E-5×$AWC/PM_{2.5}$, R=0.95)

(Fig. 3D) although the ranges of $AWC/PM_{2.5}$ were different due to the difference in

aerosol composition. This means that the fraction of aerosol liquid water governs both

the conversion of $SO_2$ to sulfate and uptake kinetics of $SO_2$.

     It should be noted that although the SOR showed a similar RH dependence as the

$SO_2$, a deviation was observed in both Shijiazhuang and Beijing (Fig. 3). The $\gamma_{SO2}$ was

measured at a fixed temperature and initial $SO_2$ concentration. In the atmosphere, both

of them varied obviously. This might lead to the observed deviation. On the other hand,

the coexisted components such as organic aerosol and black carbon in atmospheric

particles should have complicated influence on the hygroscopicity and the phase-

change of particles. The difference between the model particles and the real ambient

aerosol particles might also partially lead to the deviations of the RH dependence

between the SOR and the $\gamma_{SO2,BET}$. In addition, it also implies that besides the reaction

in aerosol liquid phase, other reaction paths such as oxidation of $SO_2$ by gas phase

oxidants should also play an important role in sulfate formation (Duan et al., 2019).

**3.3 Influence of particle composition on AWC and sulfate formation.** Besides RH,

particle composition is another important factor to affect the AWC. According to the

ions balance (Fig. S7A), ammonia was adequate to neutralize the anions in $PM_{2.5}$, which

is consistent with the results in the literature (Wang et al., 2020b). In addition,

$(81.5\pm15.9)$ % (with the median of 87.1%) of ionic anions (nitrate, chloride, and sulfate)

were neutralized by ammonium (Fig. S7B). This means $NH_4NO_3$, $(NH_4)_2SO_4$ and

$NH_4Cl$ should be the dominant form of nitrate, sulfate, and chloride in $PM_{2.5}$. We further

reconstructed the molecular composition from the ions based on the principles of

aerosol neutralization and molecular thermodynamics (Kortelainen et al., 2017). The

molecular concentrations were estimated according to the molar ratio of $NH_4^+$-to-$SO_4^{2-}$

($R_{NH4+/SO42-}$) according to the following rules: i) if $0 < R_{NH4+/SO42-} < 1$, $NH_4^+$ existed as

the chemical forms of $H_2SO_4$ and $NH_4HSO_4$. ii) $1 < R_{NH4+/SO42-} < 2$, $NH_4^+$ existed as

$(NH_4)_2SO_4$ and $NH_4HSO_4$. iii) if $R_{NH4+/SO42-} > 2$, then the fraction $NH_4^+$ corresponding

to twice the amount of $SO_4^{2-}$ existed as $(NH_4)_2SO_4$ and the remaining fraction of $NH_4^+$

was associated with $NO_3^-$ and $Cl^-$. iv) the rest of $NO_3^-$, which was not neutralized by





$NH_4^+$ was from $NaNO_3$. Figure 5A and B show the variation of the molecular

composition with RH in Shijiazhuang. Obviously, $NH_4NO_3$ and $(NH_4)_2SO_4$ were the

major molecular components. Both of them showed upward trend as the RH increased.

In particular, the fraction of $NH_4NO_3$ increased gradually from ~10 % to ~50% when

the RH increased from ~30% to 90%. Correspondingly, the fraction of $(NH_4)_2SO_4$

decreased as the RH increased.

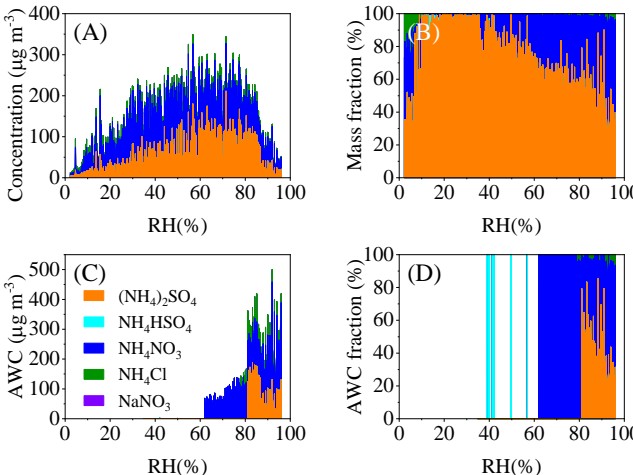

Fig. 5. Variations of (A) and (B) molecular composition of water-soluble ions, and (C)

and (D) the corresponding contributions to AWC with RH in Shijiazhuang.

     It should be noted that the deliquescence RH (DRH) of $NH_4NO_3$ (61.8 %) (Onasch

et al., 1999) is lower than those of $NH_4Cl$ (78%) (Hu et al., 2011) and $(NH_4)_2SO_4$ (80 %)

(Lightstone et al., 2000). This means ammonium nitrate should the major contributor

to the AWC compared with sulfate and chloride when both the concentrations and

hygroscopicity were taken into consideration. We further calculated the AWC attributed

to the individual molecular component. The deliquescence curve of each salt was

calculated at 298.5 K using the E-AIM model (Clegg et al., 1998). Then, the AWC was



calculated with the mass of the salt and the mass-based growth factor at the corresponding RH. As shown in Fig. 5C, $NH_4NO_3$ and $(NH_4)_2SO_4$ were the major contributors to the AWC. Especially, $NH_4NO_3$ dominated the AWC when the RH ranged from 60% to 80%, in which the SOR and the $\gamma_{SO2}$ were very sensitive to RH. These

results suggest that $NH_4NO_3$ should be the most important mediator to AWC, subsequently, the uptake of $SO_2$ in the transition regime of RH in Fig. 3A. In previous work, it has been found that $SO_2$ oxidation can be promoted by particulate nitrate through the accumulation of proton (Zhang et al., 2019) and the formation of $NO^+NO_3^-$ (Kong et al., 2014). Our results further showed the importance of $NH_4NO_3$ in the AWC,

which possibly determines the phase state of particles, subsequently, the uptake kinetics of $SO_2$ and the SOR as discussed above. To further confirm the role of $NH_4NO_3$ in the uptake of $SO_2$, uptake experiment of $SO_2$ on pure dust has been carried out at 2% and 80% RH, respectively. The corresponding $\gamma_{SO2,BET}$ was $1.10\pm1.05\times10^{-7}$ and $1.66\pm0.28\times10^{-7}$ on pure dust sample in the presence of $NH_3$ and $NO_2$. However, as

discussed above, it was 0 and $1.12\pm0.15\times10^{-5}$ on dust internally mixed with 33 % $NH_4NO_3$. This directly confirmed the role of $NH_4NO_3$ in $SO_2$ uptake via aerosol liquid water.

Figure S8 shows the dependencies of the AWC/$PM_{2.5}$ and SOR on the fraction of the individual molecular component. Both the AWC/$PM_{2.5}$ and SOR statistically

increased as the fraction of $NH_4NO_3$ increased (Fig. S8A and D). A weak increase followed by a decrease was observed for the AWC/$PM_{2.5}$ as the fraction of $(NH_4)_2SO_4$ increased, while a negative correlation between the AWC/$PM_{2.5}$ and the fraction of



NH$_4$Cl was observed. It did so for the SOR and the fraction of NH$_4$Cl. These phenomena were overall consistent with the sequence of their hygroscopicity. In addition, chloride

is a primary pollutant mainly from coal combustion and biomass burning (Bi et al., 2019). Besides chloride, other primary particles from combustion such as soot, which were not accounted for in this work, might also decrease the uptake capability of water, subsequently, be unfavorable for SO$_2$ uptake.

To assess the relative importance of sulfate and nitrate (the major SNA component)

to AWC, the sensitivity of their fraction to AWC in Shijiazhuang was tested using the ISOPRRIA II model and shown in Fig. S9. The base case means the AWC was calculated using the measured concentration of the ions. Then, we reduced the fraction of NH$_4$NO$_3$ or (NH$_4$)$_2$SO$_4$ from 0 to 80 % individually compared with the base case. Figure S9A shows the time series of the calculated AWC after reducing 50 % of

NH$_4$NO$_3$ or (NH$_4$)$_2$SO$_4$. Reduction of either NH$_4$NO$_3$ or (NH$_4$)$_2$SO$_4$ resulted into obvious decrease of AWC during pollution events. In most cases, the reduction amplitude of AWC was larger when reducing 50 % of NH$_4$NO$_3$ than (NH$_4$)$_2$SO$_4$. Figure S9B shows the mean ratio of AWC at a certain reduction fraction of NH$_4$NO$_3$ or (NH$_4$)$_2$SO$_4$ to that under the base case. When NH$_4$NO$_3$ was reduced from 0 % to 80 %,

the AWC linearly reduced from 0 % to 61.1±0.1 % with a slope of 0.48%. It varied from 0 % to 66.0±0.2 % for (NH$_4$)$_2$SO$_4$ (with a slope of 0.41%). This means that the AWC is more sensitive to the fraction of NH$_4$NO$_3$ than (NH$_4$)$_2$SO$_4$ in Shijiazhuang. This also implies the importance of NH$_4$NO$_3$ in the observed high AWC in haze days. On the other hand, reducing 10 % of NH$_4$NO$_3$ can lead to a reduction of 5.2±1.0% AWC during



haze days. Subsequently, we can roughly estimate that the SOR might be reduced by

~4 % through a linear interpolation according to the equation of the SOR and the

AWC/PM$_{2.5}$ (SOR = 0.15 + 0.40 × AWC/PM$_{2.5}$) fitted in Fig. 3C. This means reduction

of NOx and NH$_3$ should lead to additional reduction of particulate sulfate.

**3.4 Influence of other factors on sulfate formation.** Several studies have proposed

out that NO$_2$ can promote the oxidation of SO$_2$ on particle surfaces and in aqueous

phase. For example, laboratory studies have found that ppm level of NO$_2$ can promote

sulfate formation on the surface of dust through NO$^+$NO$_3^-$ which is disproportionated

from N$_2$O$_4$ intermediate (He et al., 2014;Liu et al., 2012;Ma et al., 2008), or ppm level

of NO$_2$ can promote the oxidation of SO$_2$ in the deliquesced oxalic acid (Wang et al.,

2016). This is supported by the evidence that high fraction of sulfate in PM$_{2.5}$ is

positively correlated with NO$_2$ concentration (Xie et al., 2015) and high PM$_{2.5}$

concentration is accompanied with high ratio of NO$_2$/SO$_2$ in several case studies (He et

al., 2014). The importance of the SO$_2$ oxidation by NO$_2$ in aqueous phase has also been

confirmed in modeling studies (Cheng et al., 2016;Xue et al., 2016). However, this

reaction path is still under debate because of the following reasons: 1) The

concentration of NO$_2$ in laboratory studies was about 2 orders of magnitude higher than

that in ambient air. This will affect the surface concentration of the intermediate (N$_2$O$_4$)

and the concentration of solved NO$_2$ in aqueous phase. 2) The dissolved NO$_2$

concentration is highly sensitive to pH. The pH value in aerosol was 5.6-6.2 estimated

in modeling study (Cheng et al., 2016). However, a recent work found that it varied

from 3.8 to 4.5 at RH > 30 % and showed a moderate acidity because of the



thermodynamic equilibrium between $NH_4^+$ and $NH_3$ (Ding et al., 2019). 3) The relative

importance of each path depends on the concentration of the relevant pollutants

including $H_2O_2$ and TMI (Liu et al., 2020a). Therefore, it is necessary to verify the

importance of this process by long-term observation at different environments.

Figure 6 shows the 2D Kernel density graph of the sulfate fraction in soluble PM

and the SOR in Shijiazhuang as a function of the concentration of different gas phase

pollutants. It should be pointed out that the SOR or the $\gamma_{SO2}$ should be positively

correlated to $NO_2$ concentration if it can promote the conversion of $SO_2$ to sulfate or

the uptake of $SO_2$. However, both sulfate fraction and SOR were negatively correlated

with the concentration of $NO_2$ in a point view of statistics. A same trend was observed

in Beijing (Fig. S10). This is similar to recent studies that observed the opposite

correlation between SOR and NOx concentration in Sichuan Basin (Tian et al., 2019)

and in Beijing (Fang et al., 2019). This means that $NO_2$ concentration is statistically not

a determining factor for sulfate formation in the atmosphere. This is well supported by

the uptake kinetics of $SO_2$ measured using a flow tube reactor. As shown in Fig. 3A and

B, when 50±2.5 ppb of $NH_3$ presenting in the reactant gases, no difference was

observable about the $\gamma_{SO2,BET}$ between in the presence (read squares) and absence of

100±2.5 ppb of $NO_2$ (white triangles). This is consistent with these previous studies that

found $NO_2$ having no influence on $SO_2$ uptake when $NH_3$ was abundant in the

atmosphere (Wu et al., 2019;Wang et al., 2021). In addition, it is consistent with the

fact that $H_2O_2$ dominated the oxidation of S(IV) in aerosol liquid water when RH was

higher than 60% (Fig. 4A).



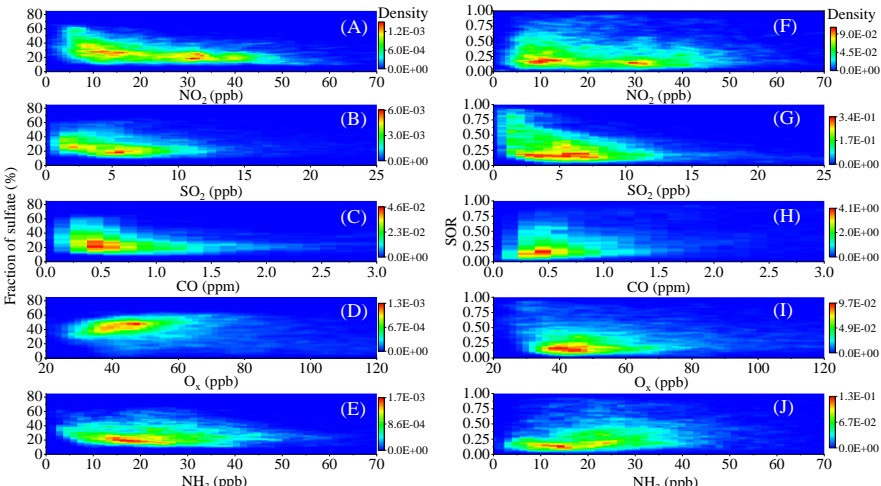

Fig. 6. Dependence of the sulfate fraction in soluble PM and the SOR on gaseous

pollutant concentration in Shijiazhuang.

Both the fraction of sulfate and the SOR in Shijiazhuang statistically decreased as

a function of $SO_2$ and CO concentration, respectively (Fig. 6B, C, G and H). This might

be explained by the high concentration of primary aerosol components when pollution

events occurred with high concentration of primary gas phase pollutants. However, the

fraction of sulfate increased as a function of $O_x$ (Fig. 6D). When the $O_x$ concentration

was greater than 50 ppb, the SOR slightly increased with the $O_x$ concentration (Fig. 6I).

A more obvious positive dependence of sulfate fraction on $O_x$ concentration was

observed in Beijing (Fig. S6D). This means oxidation capacity also plays an important

role in sulfate formation, especially in Beijing. This is consistent with the recent finding

that $O_3$ plays an important role in $SO_2$ oxidation at different locations (Fang et al.,

2019;Tian et al., 2019;Duan et al., 2019). As shown in Fig. 6J, the SOR positively

correlated with the concentration of $NH_3$ in Shijiazhuang. This means that $NH_3$ can



promote the conversion of $SO_2$ to sulfate. This is well in agreement with laboratory

studies that observed the promotion effect by $NH_3$ to the heterogeneous reaction of $SO_2$

on different mineral oxides (Yang et al., 2016). In addition, flow tube experiments were

also carried out by exposing the internal mixing sample (2:1 dust and $NH_4NO_3$) to 200

$\pm$ 2.5 ppb $SO_2$ in the absence of $NH_3$ and $NO_2$ at 2 % and 80 % RH, respectively. As

shown in Figure 3A and B, the $\gamma_{SO2, BET}$ was zero regardless of the reactants under dry

condition (2 % RH), while it increased to $(1.66 \pm 0.28) \times 10^{-6}$ at 80 % RH. However, it

was significantly smaller than the $\gamma_{SO2, BET}$ $((1.13 \pm 0.21) \times 10^{-5})$ in the presence of 50

$\pm$ 2.5 ppb $NH_3$ with or without 100 $\pm$ 2.5 ppb $NO_2$. These results further confirm that

$NH_3$ can promote the uptake of $SO_2$ at high RH, possible through enhancing the

solubility of $SO_2$ in water (Chen et al., 2019d;Cheng et al., 2016;Wang et al., 2016).

**4. Conclusions and atmospheric implications**.

Based on one-year of observations, we confirmed that high $PM_{2.5}$ mass concentration

in pollution events usually coincided with the high sulfate concentration, the fraction

of sulfate and the SOR in both Beijing and Shijiazhuang. In Shijiazhuang, the SOR

exponentially increased as a function of RH in the point view of statistics, which was

similar to the RH dependence of the $\gamma_{SO2}$ on the model particles containing 33%

$NH_4NO_3$ in the presence of $NH_3$. The SOR and $\gamma_{SO2}$ linearly increased as a function of

the fraction of aerosol water content in $PM_{2.5}$. The enhanced uptake coefficient of $SO_2$

at high RH after the liquid phase aerosol appeared might explain the increased SOR

because uptake of $SO_2$ was the rate determining step for the conversion of $SO_2$ to

particulate sulfate. $NH_4NO_3$ played an important role in the AWC, the phase state of



aerosol particles, subsequently, the uptake kinetics of $SO_2$ in haze days under high RH conditions.

The contribution of nitrate to $PM_{2.5}$ is increasing in China (Li et al., 2018;Tian et al., 2019) due to the intensive emissions of $NO_x$ from steel production and cement manufacturing (Wu et al., 2018;Qi et al., 2017) and the increasing $NO_x$ emissions from traffic (Liu et al., 2007;Wang et al., 2011). The mean fraction of nitrate in $PM_{2.5}$ was 21.4±12.4 % in Shijiazhuang and 15.8±13.4 % in Beijing, respectively. They were close to the reported values in $PM_{1.0}$ during the summer of Beijing (24 %) (Li et al., 2018) and in $PM_{2.5}$ during the winter of Chengdu (23.3 %) and Chongqing (17.5 %) (Tian et al., 2019). It has been found that the fraction of nitrate and ammonium usually increases as a function of $PM_{2.5}$ mass concentration (Li et al., 2018). Therefore, $NO_x$ should be an urgent air pollutant in the future in China even from the point view of its contribution to $PM_{2.5}$ mass.

As observed in this work, $NH_4NO_3$ has importance contribution to $PM_{2.5}$ mass concentration and the aerosol water content, subsequently, the phase state of particles in the RH range of 60-80%. Reduction of $NO_x$ emissions should lead to decrease in $NH_4NO_3$ concentration, subsequently, the AWC during serve pollution events. This will lead to an additional reduction of $SO_2$ uptake and the formation of particulate sulfate through aqueous reactions. Based on our rough estimation, 4 % of sulfate might be reduced due to aqueous reaction in Shijiazhuang if the mass concentration of $NH_4NO_3$ was reduced by 10 %. More work is required to quantitatively assess the contribution of nitrate to sulfate formation from aqueous reactions in the future. It should be noted



that ozone pollution becomes more and more important in China (Chen et al.,
2019e;Ziemke et al., 2019). This requires to harmoniously reduce $NO_x$ and volatile

organic compounds in the near future. It is also important to take actions on $NH_3$

emission control in the future as $NH_3$ can significantly promote the uptake of $SO_2$ in

liquid phase aerosol.

*Data availability.* The experimental data are available upon request to the

corresponding authors.

*Supplement.* The supplement related to this article is available online at:

*Author contributions.* YoL and XB designed the experiments. YoL and YuL wrote the

paper. ZF, FZ, YZ, XF, CY, BC, YW, WD, and JC carried out measurements at BUCT.

XB and TJ carried out measurements at SJZ. YG, YZ, and YoL carried out flow tube

experiments. PL, YM, and YoL performed sulfate formation calculations. YuL, FB, TP,

YM, HH, and MK revised the paper.

*Acknowledgments.* The research was financially supported by the National Natural

Science Foundation of China (92044301), the Ministry of Science and Technology of

the People's Republic of China (2019YFC0214701), Academy of Finland via Center of

Excellence in Atmospheric Sciences (272041, 316114, and 315203, 1307537) and

European Research Council via ATM-GTP 266 (742206), and via ERA-NET-Cofund





through SMart URBan Solutions for air quality, disasters and city growth

(SMURBS/ERA-PLANET), the Strategic Priority Research Program of Chinese

Academy of Sciences and Beijing University of Chemical Technology.

*Competing interests.* The authors declare that they have no conflict of interest.

manuscript.

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
