# Peer review of "Ammonium nitrate promotes sulfate formation through uptake kinetic regime"

_Atmospheric Chemistry and Physics, 2021_

## Referee Comment (RC2)

**Review of "Ammonium nitrate promotes sulfate formation through uptake kinetic regime " by Yongchun Liu et al. (2021)**

Very high particulate matter (PM) concentration levels are still a serious air quality and health issue in the North China Plain (NCP) that is strongly connected to formation of secondary inorganic (SIA) components such as sulfate. The formation pathway responsible for particulate sulfate in China are still highly uncertain and under debate. In the submitted manuscript, the authors present (i) a statistical analysis of long-term field observation data of two sites in the North China Plain investigating the formation mechanism of particulate sulfate and (ii) results from conducted flow tube experiments on the reactive uptake of $SO_2$ which supported the field data analyses. The analysis focuses on the impact of (1) the aerosol liquid water content (ALWC), (2) the particle composition and (3) other factors such as the concentration of important oxidants for the sulfate formation.

In my opinion, the paper under discussion is well structured, contains interesting information on an important topic of atmospheric chemistry and provides crucial implications on the formation particulate sulfate under polluted conditions in China.

However, the paper in its present forms need major revision. After addressing my comments/questions/suggestions given below, this paper might be suitable for publication in ACP.

**General comments:**

(1) Although the paper includes already a comprehensive analysis of important factors influencing the sulfate formation, aerosol acidity as one of the driving parameters for the sulfate formation and the partitioning of semi-volatile gases is hardly discussed in the paper. The paper mentions the importance of the pH only in a few places. However, the study has applied two thermodynamic models (ISOROPIA II and E-AIM) that calculate acidity. Therefore, I'm puzzled why this provided information was not used in the statistical analysis. The authors should discuss the role of acidity in an additional subsection in the revised manuscript. This would substantially improve the manuscript and the interpretation of the field data.

(2) My second major concern is about drying procedure of the dust and ammonium nitrate ($NH_4NO_3$) mixture (line 166 -170).
Have the authors checked, e.g. by an IC analysis, that there is still $NH_4NO_3$ left after their drying procedure? It is well-known that $NH_4NO_3$ is semi-volatile and its volatilization strongly depends on the temperature (see e.g., Schaap et al. (2004) and references therein). At higher temperatures, $NH_4NO_3$ evaporates. So, my concern is that no $NH_4NO_3$ left after drying at 393K (120°C). If this is the case, then the interpretation of the uptake experiments needs to be fully revised. Please provide some information on how much $NH_4NO_3$ is left after drying.

Schaap, M., et al. (2004). Artefacts in the sampling of nitrate studied in the "INTERCOMP" campaigns of EUROTRAC-AEROSOL Atmos. Environ., 38, 6487-6496, https://doi.org/10.1016/j.atmosenv.2004.08.026.

(3) In Section 3.3, the influence of the particle composition on the ALWC and sulfate formation is discussed. The fractions of the major salts were calculated by E-AIM and the deliquescence RH (DRH) of different salts are considered for the discussion. I was surprised to see no contribution of ammonium-hydrogensulfate, $(NH_4)HSO_4$ at lower RH conditions in Figure 5 and no discussion of it in the text. $(NH_4)HSO_4$ is characterized by a much lower DRH (see Li et al. (2017) and references therein) than $NH_4NO_3$. Considering this information will surely change the discussion in this section. Comparing Figure 5A and 5B, I don't understand why there is almost 100% mass fraction of $(NH_4)_2SO_4$ at 10%≤RH≤30%, but Figure 5A shows concentrations of both $NH_4NO_3$ and $(NH_4)_2SO_4$ (surprisingly no $(NH_4)HSO_4$ here!). Based on my concerns, I expect that section 3.3. and its conclusions need to be strongly revised.

Y.-J. Li, et al. (2017) Rebounding hygroscopic inorganic aerosol particles: Liquids, gels, and hydrates, Aerosol Science and Technology, 51:3, 388-396. http://dx.doi.org/10.1080/02786826.2016.1263384.

**Further Comments/Questions/Suggestions:**

Page2 line 24-25: "This implies an enhanced formation rate of $SO_4^{2-}$ in the ambient air". However, the smaller decrease in particulate sulfate can also be caused by a changed oxidation budget (increasing ozone budget) and/or because the sulfate formation in China is not a $SO_2$-limited process but rather an uptake or oxidant-limited process.

Page2 line 28: Define SOR.

Page2 line 33: Replace "in particle-phase" by "in the particle phase".

Page2 line 29: "…transition of particle phase" means "change of phase state"?

Page2 line 29: Remove "kinetics".

Page3 line 42: "(WHO, 2013;Lelieveld et al., 2015)" Please revise your citation style here and throughout the manuscript, and insert a space between the different references (after the semicolon).

Page3 line 52: The abbreviation "SNA" is quite unusual to me. Instead, I would recommend to use the abbreviation "SIA (Secondary inorganic aerosol)" that is more commonly used or remove SNA as its only used three times in the manuscript.

Page3 line 58-60: "However, the decrease rate of particulate $SO_4^{2-}$ concentration (Lang et al., 2017;Li et al., 2017) is much smaller than $SO_2$ (Lang et al., 2017;Zhang et al., 2020)." The statement is vague, so, please provide some numbers.

Page4 line 58-60: Please include the more recent reference of Liu et al. (2021).

Liu, T., A. W. H. Chan, and J. P. D. Abbatt (2021), Multiphase Oxidation of Sulfur Dioxide in Aerosol Particles: Implications for Sulfate Formation in Polluted Environments, Environ. Sci. Technol., 55(8), 4227-4242. https://doi.org/10.1021/acs.est.0c06496.

Page4 line 66: Add "in the gas phase and subsequent uptake onto particles." at the end of the sentence. Here, it should be also mentioned that the OH pathway is the dominant gas-phase oxidation pathway.

Page4 line 66: "~54.2 %" Please, provide only relevant decimal places.

Page4 line 71 and 75: Why is the $NO_2$ oxidation pathway mentioned twice?

Page4 line 71: Please include the recent references of Liu et al. (2021) and Ye et al. (2021) for the $H_2O_2$ oxidation pathway.

Liu, T., et al. (2021), Multiphase Oxidation of Sulfur Dioxide in Aerosol Particles: Implications for Sulfate Formation in Polluted Environments, Environ. Sci. Technol., 55(8), 4227-4242. https://doi.org/10.1021/acs.est.0c06496.

Ye, C., et al. (2021), Particle-Phase Photoreactions of HULIS and TMIs Establish a Strong Source of $H_2O_2$ and Particulate Sulfate in the Winter North China Plain, Environ. Sci. Technol. https://doi.org/10.1021/acs.est.1c00561.

Page4 line 75: Please include the work of Spindler et al. (2003) as this showed much lower rate constants compared to Clifton et al. (1988) and Lee and Schwartz (1983).

Spindler, G., et al. (2003), Wet annular denuder measurements of nitrous acid: laboratory study of the artefact reaction of $NO_2$ with S(IV) in aqueous solution and comparison with field measurements, Atmos. Environ., 37(19), 2643-2662, https://doi.org/10.1016/S1352-2310(03)00209-7.  .

Clifton, C. L., et al. (1988), Rate constant for the reaction of nitrogen dioxide with sulfur(IV) over the pH range 5.3-13, Environ. Sci. Technol., 22(5), 586-589. https://doi.org/10.1021/es00170a018.

Lee, Y.-N., and S. E. Schwartz (1983), Kinetics of Oxidation of Aqueous Sulfur(IV) by Nitrogen Dioxide, in Precipitation Scavenging, Dry Deposition, and Resuspension. Volume 1: Precipitation Scavenging, edited by H. R. Pruppacher, R. G. Semonin and W. G. Slinn, pp. 453-470, Elsevier, New York, Amsterdam, Oxford.

Page4 line 79: Insert "the" after "pathways to".

Page4 line 82: "an observe based" should be "observation-based"

Page5 line 83: Delete "simulation".

Page5 line 83: Add "Gas-phase" before "Oxidation".

Page5 line 84: Replace "BTH" by "the Beijing-Tianjin-Hebei province". The abbreviation is only used here once.

Page5 line 85: "simulations"

Page5 line 87: Cite also Ye et al. (2021) here.

Ye, C., et al. (2021), Particle-Phase Photoreactions of HULIS and TMIs Establish a Strong Source of

H$_2$O$_2$ and Particulate Sulfate in the Winter North China Plain, Environ. Sci. Technol. https://doi.org/10.1021/acs.est.1c00561

Page5 line 86-89: In this discussion, it would be appropriate to include results from isotope measurements and their interpretations on the key oxidation pathways. The results of conducted isotope measurements should also be included at other places in the manuscript. They can support the findings of the current study.

Page5 line 94: "what the control factors are from gas-phase SO$_2$ to particle-phase sulfate" does not sound good. Maybe better: "what are the controlling factors of the S(IV)-to-S(VI) conversion in the gas phase."

Page5 line 95-97: These two sentences here do not fit, or a transition sentence is missing.

Page5 line 100: "… statistically investigated to identify the controlling factors." Subsequently, the different sensitivity investigations should be briefly mentioned. The uptake investigations are missing in this paragraph.

Page5 line 101: Replace "mainly" by "strongly".

Page6 line 106: Remove "Lat." and "Lon.". Instead, add "N" and "E" behind the numbers, respectively.

Page6 line 120: "Particle-phase"

Page6 line 120: Measured Fe and Mn concentrations are total metal concentrations or water-soluble concentrations?

Page7 line 133: Define "IE".

Page7 line 143: "influence of RH" and "the uptake coefficient ($\gamma_{SO2}$)"

Page7 line 146: "gas-phase"

Page8 line 153: "gas-phase"

Page8 line 155: "BET" must already be defined here for the first time, and not only in line 188.

Page9 line 183: "we aimed to understand"

Page10 line 203-205: Why haven't the authors applied more simple approaches to calculate or estimate the pH. Please see Pye et al. (2020) and proxy approaches therein.

Pye, H. O. T., et al. (2020), The acidity of atmospheric particles and clouds, Atmos. Chem. Phys., 20(8), 4809-4888. https://doi.org/10.5194/acp-20-4809-2020.

Page10 line 207-212: Please, provide the references for all kinetic rate expressions given. Furthermore, the authors should look at two reviews published recently that have evaluated kinetic data on sulfur oxidation (Liu et al. (2021); Tilgner et al. (2021, under review in ACPD)). Are the applied kinetic data in agreement with their recommended values?

Liu, T., et al. (2021), Multiphase Oxidation of Sulfur Dioxide in Aerosol Particles: Implications for Sulfate Formation in Polluted Environments, Environ. Sci. Technol., 55(8), 4227-4242. https://doi.org/10.1021/acs.est.0c06496    .

Tilgner, A., et al. (2021), Acidity and the multiphase chemistry of atmospheric aqueous particles and clouds, Atmos. Chem. Phys. Discuss., https://doi.org/10.5194/acp-2021-58, in review, 2021.

Page11 line 215: "8:7" should be "8.7"

Page11 line 219: Please, cite the references for all Henry's law constants.

Page11 line 219-221: Where can I find the derived $H_2O_2$ concentrations? Do they fit to measurements in the NCP, see e.g. Ye et al. (2018)?

Ye, C., et al. (2018), High $H_2O_2$ concentrations observed during haze periods during the winter in Beijing: Importance of $H_2O_2$ oxidation in sulfate formation, Environ. Sci. Tech. Let., 5(12), 757-763, https://doi.org/10.1021/acs.estlett.8b00579.

Page11 line 235-236: Please, provide only relevant decimal places.

Page12 line 238: Remove "well".

Page12 line 238: "larger population of heavy industries" sounds bad. Do you mean "larger density of heavy industries"?

Page12 line 240: "than in Beijing".

Page12 Fig1: The legend is not well placed.

Page13 line 261: Another consequence of "the increased traffic emissions in Beijing", i.e. higher NOx emissions, is that the concentrations of ozone are elevated in Beijing. This should be mentioned!

Page13 line 272: Better say that the Shijiazhuang site is more influenced by primary emissions.

Page13 line 274: Please clarify "significantly higher". 55 ppb and 51 ppb are not significantly different!

Page14 line 275-279: I do not agree with the conclusion drawn here, because of the higher primary emissions in Shijiazhuang affecting the SOR. Perhaps other parameters are required to reach this conclusion.

Page14 line 276: "gas-phase"

Page14 line 276: "multiphase"

Page14 line 282-283: "PM2.5 mass concentration well kept pace with the high sulfate concentration" sounds bad.

Page14 line 288: "a similar".

Page15 line 288: "As shown in Fig. 2D, the high concentration of sulfate positively correlated with high RH in most cases". I'm not convinced here and it's hard to see from the Figure! Please provide a correlation coefficient.

Page18 line 352: "gas-phase".

Page19 line 368: "the uptake".

Page19 line 369: "a quick".

Page19 line 373: "metals".

Page19 line 373: "metals".

Page19 line 381: I think Fig.S5 contains important information and should be therefore part of the main manuscript.

Page23 line 451: Replace "with" by "as a function of".

Page23 line 457-458: Please revise the Figure caption and describe in more detail what is shown in the different items.

Page23 line 459-461: Please see e.g. Li et al. (2017) for more recent DRH values incl. other salts. Why $(NH_4)HSO_4$ is not listed here which has a lower DRH than $(NH_4)NO_3$? Therefore, the following conclusion ("…ammonium nitrate should the major contributor to the AWC compared with sulfate and chloride…") can be wrong and the subsequent discussion should be revised.

Li, Y. J., et al. (2016), Rebounding hygroscopic inorganic aerosol particles: Liquids, gels, and hydrates, Aerosol Sci. Technol., 51(3), 388-396. https://doi.org/10.1080/02786826.2016.1263384.

Page23 line 465: Here, the E-AIM model is mentioned for the first time. Why not in Section 2? Would it be possible to use only E-AIM or ISOROPIA in the present study?

Page26 line 524-535: A recently submitted review by Tilgner et al. (2021, under review in ACPD) has outlined that the reaction rate constant of the $NO_2$ reaction with dissolved S(IV) by Clifton et al. (1988) is far too high and that studies by Spindler et al. (2003) showed much lower values. This fact should be also reflected in the discussion here.

Clifton, C. L., et al. (1988), Rate constant for the reaction of nitrogen dioxide with sulfur(IV) over the pH range 5.3-13, Environ. Sci. Technol., 22(5), 586-589. https://doi.org/10.1021/es00170a018. Spindler, G., et al. (2003), Wet annular denuder measurements of nitrous acid: laboratory study of the artefact reaction of $NO_2$ with S(IV) in aqueous solution and comparison with field measurements, Atmos. Environ., 37(19), 2643-2662, https://doi.org/10.1016/S1352-2310(03)00209-7. Tilgner, A., et al. (2021), Acidity and the multiphase chemistry of atmospheric aqueous particles and clouds, Atmos. Chem. Phys. Discuss., https://doi.org/10.5194/acp-2021-58, in review, 2021.

Page28 line Fig.6: In this Figure, it would be better to use $O_3$ instead of $O_x$, because $NO_2$ is also considered separately.

Page28 line 560: "gas-phase".

Page29 line 577-579: Here, it should be mentioned that the effective solubility of $SO_2$ can be enhanced due the increase of the aerosol pH. Furthermore, a lower acidity also promotes other oxidation processes and enables therefore higher S(VI) formation rates.

Page29 line 588: "liquid-phase".

Page32 line 643 ff: Please check again all references. The reference style is not uniform, for example the doi style.

Supporting Information (SI): The Figure captions in the SI are in parts rather brief. I strongly recommend to extend the captions, especially for complex Figures with multiple items.

---

## Author Comment (AC1)

Dear Reviewer,

We appreciate your careful consideration of our manuscript. We have carefully responded to all of your point-by-point comments and issues and have revised the manuscript accordingly. These revisions are described in detail below.

**Reviewer 1#**

The dominant formation pathway of sulfate aerosols under haze conditions is still under debate. Liu et al. investigated the formation mechanism of particulate sulfate based on a statistical analysis of long-term observations in Shijiazhuang and Beijing supported with flow tube experiments. They found that the uptake of $SO_2$ is the rate-determining step of sulfate formation. Ammonium nitrate plays an important role in sulfate formation by impacting the aerosol liquid water content and the phase state of particles. Overall, the paper is well written and the findings have important implications for understanding the sulfate chemistry under haze conditions and improving the air quality in urban environments. However, I have some concerns regarding methods and data analysis that must be addressed before the paper can be considered for publication.

**Response**: Thank you for your positive comments.

**General comments:**

Uptake kinetics of $SO_2$: the authors stated that the RDS of sulfate formation should be the uptake of $SO_2$ because the dependence of sulfate formation rate on RH is opposite to the dependences of SOR and $\gamma_{SO2}$ on RH. Did the uptake of $SO_2$ refer to the mass transfer of $SO_2$ to aerosol particles? If yes, the rates of mass transfer of $SO_2$ and aqueous oxidation of S(IV) can be calculated using a resistance model (Cheng et al. 2016). According to Cheng et al. (2016), the mass transfer of $SO_2$ is not the rate-determining step.

**Response**: Thank you for your good comments and suggestion. Yes, the uptake of $SO_2$ refers to the mass transfer of $SO_2$ to aerosol particles. Fig. R1 shows the probability weighted production rate of sulfate through mass transfer (uptake) and aqueous phase oxidation of $SO_2$ in Shijiazhuang. The mass transfer of $SO_2$ to aerosol particles is the

RDS, in particular, when RH is lower than 70%. We added the production rate of sulfate through mass transfer in Fig. 4 in the revised manuscript.

[Figure]

Fig. R1. Dependence of the probability weighted production rate of sulfate through mass transfer and aqueous oxidation of S(IV) in Shijiazhuang.

Using a resistance model, Cheng et al (2016) proposed out that the mass transfer of $SO_2$ is not the RDS. It should be noted that a large mass accommodation coefficient of $SO_2$ ($\alpha = 0.11$) was used in their calculations (Cheng et al., 2016). According to the relationship between the mass accommodation coefficient ($\alpha$) and the uptake coefficient ($\gamma$) of $SO_2$ (Kulmala and Wagner, 2001), the $\alpha_{SO2}$ on particles is on the same order of the $\gamma_{SO2}$ (Fig. R2). This means that the mass accommodation of $SO_2$ on particles should be much lower than the value used by Cheng et al (2016) according to the measured uptake coefficient of $SO_2$ in this work. Therefore, the mass transfer rate coefficient ($k_{MT}$) is much lower than their value. We added a paragraph "We further calculated the production rate of sulfate through uptake of $SO_2$ (mass transfer to aerosol particles) according to,

$$\frac{d[SO_4^{2-}]}{dt} = 3600 \cdot \frac{96}{64} \cdot \frac{\gamma_{SO_2} A_s \omega c_{SO_2}}{4} \qquad (8)$$

where, $A_s$ is the surface area concentration of PM$_{2.5}$, $\omega$ is the mean molecular velocity of $SO_2$ and $c_{SO2}$ is the mass concentration of $SO_2$. As shown in Fig. 4C, the probability weighted production rate of sulfate through uptake of $SO_2$ (the grey line) is lower than

that through aqueous oxidation of S(IV), in particular, when RH is lower than 70%. It should be noted the mass transfer of $SO_2$ was not thought as the RDS using a large mass accommodation coefficient of $SO_2$ ($\alpha = 0.11$) (Cheng et al., 2016). According to the relationship between the mass accommodation coefficient ($\alpha$) and the uptake coefficient ($\gamma$) of $SO_2$ (Kulmala and Wagner, 2001), the $\alpha_{SO2}$ on particles is on the same order of the $\gamma_{SO2}$. This means that mass transfer rate might be greatly overestimated by Cheng et al. (2016)" in lines 432-444 in the revised manuscript.

[Figure]

Fig. R2. The relationship between the mass accommodation coefficient ($\alpha$) and the uptake coefficient ($\gamma$) of $SO_2$ for 20 nm and 150 nm particles at 1 atm and 298 K.

Flow tube experiments: (1) The wall loss of $SO_2$ on the inner surface of the outside tube and the outside surface of the sample holder was subtracted. However, the wall loss of $SO_2$ in the presence of $NH_3$ and/or $NO_2$ would be larger even in the absence of seed aerosols (Ge et al., 2019), which may lead to an overestimation of $\gamma_{SO2}$ in the presence of $NH_3$ and/or $NO_2$. Did the authors measure the $\gamma_{SO2}$ in the presence of $NH_3$ and/or $NO_2$ without the presence of seed aerosols?

**Response**: Thank you. We agree with you that the wall loss of $SO_2$ in the presence of $NH_3$ and/or $NO_2$ would be larger in the absence of seed aerosols. The $\gamma_{SO2}$ was $2\times10^{-7}$ in the presence of $NH_3$ and $NO_2$ and in the absence of seed particles. This value is significantly lower than that in the presence of particles ($\sim1\times10^{-5}$). On the other hand,

the $c_0$ and $c$ of $SO_2$ are required when we calculating the uptake coefficient. Figure R3A shows the configuration of the flow tube reactor for measuring the $c_0$, and Figure R3B shows that for measuring the $c$. Because all the inner surface of the sample tube is covered by particles, the contribution of the wall loss of $SO_2$ to the drop of the $c$ is limited (less than 3%). We added a short paragraph "It should be noted that the wall loss of $SO_2$ in the presence of $NH_3$ and/or $NO_2$ would be larger in the absence of seed aerosols. Additional control experiments in the presence of $NO_2$ and $NH_3$ demonstrate that the contribution of wall loss of $SO_2$ should be less than 3 % to the measured γ" in lines 211-214 in the revised manuscript.

[Figure]

Fig. R3. The configuration of the flow tube for measuring the $c_0$ and $c$ of $SO_2$.

(2) Can 100 ppb of $NO_2$ oxidize 190 ppb of $SO_2$ at a detectable rate in around 1 min?

The comparable $\gamma_{SO2}$ in the absence and presence of $NO_2$ may not demonstrate that $NO_2$ is not an important oxidant of $SO_2$ if the enhanced uptake of $SO_2$ in the presence of $NO_2$ is too low under the experimental conditions of the paper.

90 **Response**: Thank you so much for your good comment. The $\gamma_{SO2}$ at 80% RH was $1.7\pm0.3\times10^{-6}$ on the mixture of dust and $NH_4NO_3$ in the absence of $NH_3$ and $NO_2$ (Fig. R4 and Fig. 3). It increased to $3.7\pm0.2\times10^{-6}$ in the presence of $NO_2$ at the same RH. This means that $NO_2$ can promote $SO_2$ uptake at high RH in the absence of $NH_3$. Thus, the short reaction time should not be a problem. However, in the presence of $NH_3$, the

95 $\gamma_{SO2}$ showed no difference between with and without $NO_2$. This means that the promotion effect of $NO_2$ on $SO_2$ uptake might be too low to be detected in the presence of $NH_3$. Because $NH_3$ is abundant in North China, we think the promotion effect of $NO_2$ alone to $SO_2$ uptake should be limited in the atmosphere. In the revised manuscript, we added a paragraph "It should be pointed out that the $\gamma_{SO2}$ at 80% RH was $1.7\pm0.3\times10^{-6}$

100 on the mixture of dust and $NH_4NO_3$ in the absence of $NH_3$ and $NO_2$ (Fig. 3). It increased to $3.7\pm0.2\times10^{-6}$ in the presence of $NO_2$. This is consistent with the promotion effect of $NO_2$ for converting $SO_2$ to sulfate in the absence of $NH_3$ as observed in both a smog chamber (Wang et al., 2016) and a bubbling reactor (Chen et al., 2019d). However, the enhanced uptake of $SO_2$ induced by $NO_2$ might be too low to be measured in the

105 presence of $NH_3$. Therefore, the weak promotion effect by $NO_2$ alone cannot explain the negative correlation between the SOR and the concentration of $NO_2$ in Fig. 6F" in lines 607-615 in the revised manuscript.

[Figure]

Fig. R4. Relationship between SOR and $\gamma_{SO2,BET}$ on dust internally mixed with $NH_4NO_3$ (2:1) and RH in (A) Shijiazhuang and (B) Beijing, and the correlation of (C) SOR in Shijiazhuang and (D) $\gamma_{SO2,BET}$ with AWC/PM$_{2.5}$. The initial concentrations of $SO_2$, $NO_2$ and/or $NH_3$ in the flow tube reactor were $190 \pm 2.5$, $100 \pm 2.5$ and/or $50 \pm 2.5$ ppb, respectively. The grey lines are the fitting curves for the most probable SOR and the white lines are the fitting curves for the $\gamma_{SO2,BET}$.

Specific comments:

Lines 191-192: Did the control experiments run in the presence of $NH_3$ and $NO_2$?

**Response**: Thank you. Yes, it has been done. We added a short paragraph "It should be noted that the wall loss of $SO_2$ in the presence of $NH_3$ and/or $NO_2$ would be larger in the absence of seed aerosols. Additional control experiments in the presence of $NO_2$ and $NH_3$ demonstrate that the contribution of wall loss of $SO_2$ should be less than 3 % to the measured $\gamma$" in lines 211-214 in the revised manuscript.

Lines 368-372: The oxidation of $SO_2$ by $O_2$ on the aqueous microdroplets has been found to occur under acidic conditions (pH <3). What is the aerosol pH of the mixture of ammonium nitrate and dust?

**Response**: Thank you so much for your comment. We cannot calculate or measure the pH of the mixture of $NH_4NO_3$ and dust. The pH of deliquesced $NH_4NO_3$ is around 4.2

calculated using the ISORROPIA II model. This value is close to the literature value (pH<3.5) (Hung et al., 2018). We revised the paragraph "Another study also observed a quick formation of sulfate on the surface of aqueous microdroplets under acidic conditions (pH < 3.5) without the addition of other oxidants, which was explained by the direct interfacial electron transfer from $SO_2$ to $O_2$ on the aqueous microdroplets (Hung et al., 2018). The pH of deliquesced $NH_4NO_3$ is 4.2 calculated using the ISORROPIA II model" in lines 398-403 in the revised manuscript.

Fig 5C: The AWC was attributed to individual components using E-AIM model. Are the concentrations of the total AWC consistent with the ISORROPIA model? At RH of 60%-80%, only ammonium nitrate aerosols contributed to the AWC. Does this indicate that ammonium sulfate aerosols are effloresced and phase-separated with ammonium nitrate aerosols? Please explain why ammonium sulfate aerosols and ammonium nitrate aerosols are not in the same liquid phase.

**Response**: Thank you for your good comments. The AWC attributed to the individual salt cannot be separated from that of $PM_{2.5}$ using the ISORROPIA model. Thus, it was estimated using the reconstructed mass concentration of the salts and the growth factors. In Fig. R5 (Fig. 5C in the revised manuscript), we compared the total AWC of $PM_{2.5}$ calculated using the ISORROPIA model with the sum of the AWC attributed to the individual salt using the E-AIM model. Overall, the latter one underestimates around 13% of the AWC. This should be related to the difference in the mixing state between these two calculation methods. $NH_4NO_3$ dominates the AWC at RH of range 60-80%. However, we don't think this means $(NH_4)_2SO_4$ aerosols are effloresced and phase-separated with $NH_4NO_3$. As shown in Fig. R5, $NH_4NO_3$ explained ~70% of the AWC of $PM_{2.5}$. Thus, we think it is reasonable to draw a conclusion that $NH_4NO_3$ is the dominant contributor to AWC in the RH range of 60-80%. In the revised manuscript, we added a new paragraph "As shown in Fig. 5C, the sum of the AWC of individual salts overall underestimated around 13 % of that calculated using the ISORROPIA II model (the gray line) because the mixing state was not considered in the former method. However, we can still draw a conclusion that $NH_4NO_3$ and $(NH_4)_2SO_4$ are the major

contributors to the AWC” in lines 511-515 in the revised manuscript.

[Figure]

160

Fig. R5. Variations of (A) concentrations and (B) fractions of molecular composition of water-soluble ions, and (C) and (D) the corresponding contributions to AWC with RH in Shijiazhuang.

Lines 545-551: The authors should rule out the possibility that the enhanced uptake of $SO_2$ induced by $NO_2$ in the reaction time scale of the flow tube experiments is too low to be measured. Previous smog chamber experiments with longer reaction times have demonstrated that $NO_2$ can promote sulfate formation (Wang et al., 2016; Chen et al., 2019).

**Response**: Thank you for your suggestion. We added a paragraph as “It should be pointed out that the $\gamma_{SO2}$ at 80% RH was $1.7\pm0.3\times10^{-6}$ on the mixture of dust and $NH_4NO_3$ in the absence of $NH_3$ and $NO_2$ (Fig. 3). It increased to $3.7\pm0.2\times10^{-6}$ in the presence of $NO_2$. This is consistent with the promotion effect of $NO_2$ for converting $SO_2$ to sulfate in the absence of $NH_3$ as observed in both a smog chamber (Wang et al., 2016) and a bubbling reactor (Chen et al., 2019d). However, the enhanced uptake of $SO_2$ induced by $NO_2$ might be too low to be measured in the presence of $NH_3$. Therefore, the weak promotion effect by $NO_2$ alone cannot explain the negative correlation between the SOR and the concentration of $NO_2$ in Fig. 6F” in lines 607-615 in the revised manuscript.

180

Technical comments:

Line 28: Write out "SOR".

**Response**: Thank you. It has been defined as "sulfur oxidation ratio" in line 28 in the revised manuscript.

185

Fig 5: Variations of (A) concentrations…

**Response**: Thank you. We revised the caption as "Variations of (A) the mass concentrations and (B) the mass fractions of molecular composition in $PM_{2.5}$, (C) the estimated AWC attributed to different composition and (D) the corresponding AWC

190 fraction as a function of RH in Shijiazhuang" in the revised manuscript.

**References:**

Cheng, Y., Zheng, G., Wei, C., Mu, Q., Zheng, B., Wang, Z., Gao, M., Zhang, Q., He, K., Carmichael, G., Poschl, U., and Su, H.: Reactive nitrogen chemistry in aerosol water as a source of sulfate during

195 haze events in China, Sci. Adv., 2, https://doi.org/10.1126/sciadv.1601530, 2016.

Hung, H.-M., Hsu, M.-N., and Hoffmann, M. R.: Quantification of $SO_2$ oxidation on interfacial surfaces of acidic micro-droplets: Implication for ambient sulfate formation, Environ. Sci. Technol., 52, https://doi.org/9079-9086, 10.1021/acs.est.8b01391, 2018.

Kulmala, M., and Wagner, P. E.: Mass accommodation and uptake coefficients — a quantitative

200 comparison, J. Aerosol Sci., 32, 833-841, https://doi.org/10.1016/S0021-8502(00)00116-6, 2001.

---

## Author Comment (AC2)

Dear Reviewer,

We appreciate your careful consideration of our manuscript. We have carefully responded to all of your point-by-point comments and issues and have revised the manuscript accordingly. These revisions are described in detail below.

**Review 2#**

Very high particulate matter (PM) concentration levels are still a serious air quality and health issue in the North China Plain (NCP) that is strongly connected to formation of secondary inorganic (SIA) components such as sulfate. The formation pathway responsible for particulate sulfate in China are still highly uncertain and under debate. In the submitted manuscript, the authors present (i) a statistical analysis of long-term field observation data of two sites in the North China Plain investigating the formation mechanism of particulate sulfate and (ii) results from conducted flow tube experiments on the reactive uptake of SO2 which supported the field data analyses. The analysis focuses on the impact of (1) the aerosol liquid water content (ALWC), (2) the particle composition and (3) other factors such as the concentration of important oxidants for the sulfate formation. In my opinion, the paper under discussion is well structured, contains interesting information on an important topic of atmospheric chemistry and provides crucial implications on the formation particulate sulfate under polluted conditions in China. However, the paper in its present forms need major revision. After addressing my comments/questions/suggestions given below, this paper might be suitable for publication in ACP.

**Response**: Thank you for your positive comments.

General comments:

(1) Although the paper includes already a comprehensive analysis of important factors influencing the sulfate formation, aerosol acidity as one of the driving parameters for the sulfate formation and the partitioning of semi-volatile gases is hardly discussed in the paper. The paper mentions the importance of the pH only in a few places. However,

the study has applied two thermodynamic models (ISOROPIA II and E-AIM) that calculate acidity. Therefore, I'm puzzled why this provided information was not used in the statistical analysis. The authors should discuss the role of acidity in an additional subsection in the revised manuscript. This would substantially improve the manuscript and the interpretation of the field data.

**Response**: Thank you for your good suggestion. Aerosol acidity has complicated influences on sulfate formation. As shown in Fig. R1, when aerosol pH is lower than 4.5, the oxidation rate of S(IV) in aerosol liquid phase decreases as a function of pH because oxidation of S(IV) by transition metals is the dominant path, which is negatively dependent on aerosol pH. However, it increases as a function of aerosol pH when the pH is higher than 4.5 because the solubility and effective Henry's law constant of $SO_2$ are positively dependent on pH (Cheng et al., 2016;Liu et al., 2021;Liu et al., 2020). We added a short paragraph "Aerosol acidity is one of important factors affecting the sulfate formation and the partitioning of semi-volatile gases in the atmosphere (Liu et al., 2021). As shown in Fig. S11, when aerosol pH is lower than 4.5, the oxidation rate of S(IV) in aerosol liquid phase decreases as a function of pH because the oxidation of S(IV) by transition metals is the dominant path and is negatively dependent on aerosol pH. However, the oxidation rate of S(IV) increases when the aerosol pH is higher than 4.5. This can be explained by the fact that the solubility and effective Henry's law constant of $SO_2$ are positively dependent on pH (Cheng et al., 2016;Liu et al., 2021;Liu et al., 2020), which is consistent with the promotion effect of sulfate formation by $NH_3$" in lines 642-650 in the revised manuscript.

[Figure]

Fig. R1. The dependence of the oxidation rate of S(IV) in aerosol liquid phase on aerosol pH in Shijiazhuang. The white circles are the probability weighted values.

55

(2) My second major concern is about drying procedure of the dust and ammonium nitrate ($NH_4NO_3$) mixture (line 166 -170). Have the authors checked, e.g. by an IC analysis, that there is still $NH_4NO_3$ left after their drying procedure? It is well-known that $NH_4NO_3$ is semi-volatile and its volatilization strongly depends on the temperature

60   (see e.g., Schaap et al. (2004) and references therein). At higher temperatures, $NH_4NO_3$ evaporates. So, my concern is that no $NH_4NO_3$ left after drying at 393K (120°C). If this is the case, then the interpretation of the uptake experiments needs to be fully revised. Please provide some information on how much $NH_4NO_3$ is left after drying.

Schaap, M., et al. (2004). Artefacts in the sampling of nitrate studied in the

65   "INTERCOMP" campaigns of EUROTRAC-AEROSOL Atmos. Environ., 38, 6487-6496, https://doi.org/10.1016/j.atmosenv.2004.08.026.

**Response**: Thank you so much for your good comment. Yes, we checked the composition of the mixture of dust and $NH_4NO_3$ with an IC. 49.7 % of $NH_4NO_3$ was still remained in the mixture.

70        On the other hand, we compared the $\gamma_{SO2}$ on different samples in the presence of

$NO_2$ and $NH_3$ at 80% RH. As shown in Table R1, the $\gamma_{SO2}$ on the mixture of dust and $NH_4NO_3$ is comparable with that on the mixture of dust and $NaNO_3$. In addition, the $\gamma_{SO2}$ on the mixture samples containing nitrate is significantly higher than that on the pure dust sample, which is comparable with that on $\alpha$-$Fe_2O_3$ and $\gamma$-$Al_2O_3$ reported in our previous work (Yang et al., 2019). This further supported the IC results.

Table R1. The uptake coefficient of $SO_2$ on different samples at RH 80%

| Samples | Atmosphere | $\gamma_{SO2}$ ($10^{-5}$) |
|---------|------------|---------------------------|
| Dust | $SO_2 + NO_2 + NH_3$ | $0.030 \pm 0.004$ |
| Dust+$NaNO_3$ | $SO_2 + NO_2 + NH_3$ | $1.23 \pm 0.15$ |
| Dust+$NH_4NO_3$ | $SO_2 + NO_2 + NH_3$ | $1.12 \pm 0.13$ |

We added a sentence "$NH_4NO_3$ in the mixture was further confirmed using an Ion Chromatograph ($\Omega$ Metrohm 940, Applikon Analytical B.V., Netherlands). Around 50 % of $NH_4NO_3$ remained in the mixture due to evaporation." in lines 186-189 in the revised manuscript.

(3) In Section 3.3, the influence of the particle composition on the ALWC and sulfate formation is discussed. The fractions of the major salts were calculated by E-AIM and the deliquescence RH (DRH) of different salts are considered for the discussion. I was surprised to see no contribution of ammonium-hydrogensulfate, $(NH_4)HSO_4$ at lower RH conditions in Figure 5 and no discussion of it in the text. $(NH_4)HSO_4$ is characterized by a much lower DRH (see Li et al. (2017) and references therein) than $NH_4NO_3$. Considering this information will surely change the discussion in this section. Comparing Figure 5A and 5B, I don't understand why there is almost 100% mass fraction of $(NH_4)_2SO_4$ at 10%≤RH≤30%, but Figure 5A shows concentrations of both $NH_4NO_3$ and $(NH_4)_2SO_4$ (surprisingly no $(NH_4)HSO_4$ here!). Based on my concerns, I expect that section 3.3. and its conclusions need to be strongly revised.

Y.-J. Li, et al. (2017) Rebounding hygroscopic inorganic aerosol particles: Liquids, gels, and hydrates, Aerosol Science and Technology, 51:3, 388-396

**Response**: Thank you for your good comment. We agree with you that $(NH_4)HSO_4$ has

a lower DRH than $NH_4NO_3$. However, $NH_3$ is abundant in North China to neutralize sulfuric and nitrous acids in $PM_{2.5}$. For example, the annual mean concentration of $NH_3$ was $34.5\pm18.0$ ppb in Shijiazhuang. Figure R2 shows the molar ratio of $NH_4^+/SO_4^{2-}$ ($R_{NH4+/SO42-}$) in Shijiazhuang. 98.4 % of the dataset showed the $R_{NH4+/SO42-}$ higher than 2.0, which means the corresponding $NH_4HSO_4$ concentration values were zero. For the rest data with the $R_{NH4+/SO42-}$ less than 2.0, $NH_4HSO_4$ concentrations were very low (with mean and median values of 0.12 and 0.007 $\mu g\ m^{-3}$). This is the reason why we cannot see the contribution of $NH_4HSO_4$ to $PM_{2.5}$ in Fig. 5A and B. As shown in Fig. 5D, $NH_4HSO_4$ is observable but the absolute concentration is too low to be seen in Fig. 5C.

In the revised manuscript, we added a new short paragraph "It should be noted that $(NH_4)HSO_4$ has a lower DRH than $NH_4NO_3$ (Li et al., 2017b). However, 98.4% of the data points showed the $R_{NH4+/SO42-}$ higher than 2.0 in Shijiazhuang. This means that the contribution of $(NH_4)HSO_4$ to $PM_{2.5}$ should be negligible because of the abundance of atmospheric $NH_3$ in North China" in lines 517-521 in the revised manuscript.

[Figure]

Fig. R2. Variation of the $NH_4^+/SO_4^{2-}$ ratio in Shijiazhuang

**Further Comments/Questions/Suggestions:**

115 Page2 line 24-25: "This implies an enhanced formation rate of $SO_4^{2-}$ in the ambient air". However, the smaller decrease in particulate sulfate can also be caused by a changed oxidation budget (increasing ozone budget) and/or because the sulfate formation in China is not a $SO_2$-limited process but rather an uptake or oxidant-limited process.

**Response:** Thank you. We agree with you that increases in oxidation budget or

120 oxidation-limited process and uptake process can lead to the observed smaller decrease rate of $SO_4^{2-}$ than $SO_2$ in China. We think this is not conflict with our statement "This implies an enhanced formation rate of $SO_4^{2-}$ in the ambient air, and the mechanism is still under debate".

125 Page2 line 28: Define SOR.

**Response**: Thank you. It has been defined "sulfur oxidation ratio (SOR)" in line 28 in the revised manuscript.

Page2 line 33: Replace "in particle-phase" by "in the particle phase".

130 **Response**: Thank you. It has been corrected in line 33 in the revised manuscript.

Page2 line 29: "…transition of particle phase" means "change of phase state"?

**Response**: Thank you. It has been corrected in lines 34-35 in the revised manuscript.

135 Page2 line 29: Remove "kinetics".

**Response**: Thank you. We revised it to "Our results" in line 32 in the revised manuscript.

Page3 line 42: "(WHO, 2013;Lelieveld et al., 2015)" Please revise your citation style

140 here and throughout the manuscript, and insert a space between the different references (after the semicolon).

**Response**: Thank you. It has been corrected in line 49 and other places throughout the manuscript.

Page3 line 52: The abbreviation "SNA" is quite unusual to me. Instead, I would recommend to use the abbreviation "SIA (Secondary inorganic aerosol)" that is more commonly used or remove SNA as its only used three times in the manuscript.

**Response**: Thank you. It has been replaced with "Secondary inorganic aerosol (SIA)" in lines 51 and 54 in the revised manuscript.

Page3 line 58-60: "However, the decrease rate of particulate $SO_4^{2-}$ concentration (Lang et al., 2017;Li et al., 2017) is much smaller than $SO_2$ (Lang et al., 2017;Zhang et al., 2020)." The statement is vague, so, please provide some numbers.

**Response**: Thank you for your suggestion. We added a new sentence here "For example, the annual mean concentration of $SO_4^{2-}$ decreased by 0.1 $\mu g\ m^{-3}\ year^{-1}$ from 2000 to 2013, followed by 1.9 $\mu g\ m^{-3}\ year^{-1}$ from 2013 to 2015 in Beijing, while it decreased by 3.8 $\mu g\ m^{-3}\ year^{-1}$ for $SO_2$ (Lang et al., 2017)" in lines 60-63 in the revised manuscript.

Page4 line 58-60: Please include the more recent reference of Liu et al. (2021).

Liu, T., A. W. H. Chan, and J. P. D. Abbatt (2021), Multiphase Oxidation of Sulfur Dioxide in Aerosol Particles: Implications for Sulfate Formation in Polluted Environments, Environ. Sci. Technol., 55(8), 4227-4242. https://doi.org/10.1021/acs.est.0c06496.

**Response**: Thank you so much. It has been included in line 60 in the revised manuscript.

Page4 line 66: Add "in the gas phase and subsequent uptake onto particles." at the end of the sentence. Here, it should be also mentioned that the OH pathway is the dominant gas-phase oxidation pathway.

**Response**: Thank you so much. This sentence has been revised "Particulate $SO_4^{2-}$ can be formed through homogeneous oxidation of $SO_2$ by hydroxyl radicals (OH) and Stabilized Criegee Intermediates (SCIs) in the gas phase and subsequent uptake onto particles, while the OH pathway is the dominant gas-phase oxidation pathway" in lines

 in the revised manuscript.

Page4 line 66: "~54.2 %" Please, provide only relevant decimal places.

**Response**: Thank you. It has been corrected in line 72 in the revised manuscript.

Page4 line 71 and 75: Why is the $NO_2$ oxidation pathway mentioned twice?

**Response**: Thank you. We moved that in line 76 to line 80 in the revised manuscript.

Page4 line 71: Please include the recent references of Liu et al. (2021) and Ye et al. (2021) for the $H_2O_2$ oxidation pathway.

Liu, T., et al. (2021), Multiphase Oxidation of Sulfur Dioxide in Aerosol Particles: Implications for Sulfate Formation in Polluted Environments, Environ. Sci. Technol., 55(8), 4227-4242. https://doi.org/10.1021/acs.est.0c06496.

Ye, C., et al. (2021), Particle-Phase Photoreactions of HULIS and TMIs Establish a Strong Source of H2O2 and Particulate Sulfate in the Winter North China Plain, Environ. Sci. Technol. https://doi.org/10.1021/acs.est.1c00561.

**Response**: Thank you so much. It has been included in line 77 in the revised manuscript.

Page4 line 75: Please include the work of Spindler et al. (2003) as this showed much lower rate constants compared to Clifton et al. (1988) and Lee and Schwartz (1983).

Spindler, G., et al. (2003), Wet annular denuder measurements of nitrous acid: laboratory study of the artefact reaction of $NO_2$ with S(IV) in aqueous solution and comparison with field measurements, Atmos. Environ., 37(19), 2643-2662, https://doi.org/10.1016/S1352-2310(03)00209-7.

Clifton, C. L., et al. (1988), Rate constant for the reaction of nitrogen dioxide with sulfur(IV) over the pH range 5.3-13, Environ. Sci. Technol., 22(5), 586-589. https://doi.org/10.1021/es00170a018.

Lee, Y.-N., and S. E. Schwartz (1983), Kinetics of Oxidation of Aqueous Sulfur(IV) by Nitrogen Dioxide, in Precipitation Scavenging, Dry Deposition, and Resuspension.

Volume 1: Precipitation Scavenging, edited by H. R. Pruppacher, R. G. Semonin and

W. G. Slinn, pp. 453-470, Elsevier, New York, Amsterdam, Oxford.

**Response**: Thank you so much. It has been included in line 81 in the revised manuscript.

205

Page4 line 79: Insert "the" after "pathways to".

**Response**: Thank you. It has been corrected in line 85 in the revised manuscript.

Page4 line 82: "an observe based" should be "observation-based"

210 **Response**: Thank you. It has been corrected in line 88 in the revised manuscript.

Page5 line 83: Delete "simulation".

**Response**: Thank you. It has been corrected in line 88 in the revised manuscript.

215 Page5 line 83: Add "Gas-phase" before "Oxidation".

**Response**: Thank you. It has been corrected in line 89 in the revised manuscript.

Page5 line 84: Replace "BTH" by "the Beijing-Tianjin-Hebei province". The

abbreviation is only used here once.

220 **Response**: Thank you. It has been corrected in lines 89-90 in the revised manuscript.

Page5 line 85: "simulations"

**Response:** Thank you. It has been corrected in line 91 in the revised manuscript.

225 Page5 line 87: Cite also Ye et al. (2021) here.

Ye, C., et al. (2021), Particle-Phase Photoreactions of HULIS and TMIs Establish a

Strong Source of $H2O2$ and Particulate Sulfate in the Winter North China Plain,

Environ. Sci. Technol. https://doi.org/10.1021/acs.est.1c00561

**Response**: Thank you. It has been cited in line 93 in the revised manuscript.

230

Page5 line 86-89: In this discussion, it would be appropriate to include results from isotope measurements and their interpretations on the key oxidation pathways. The results of conducted isotope measurements should also be included at other places in the manuscript. They can support the findings of the current study.

**Response**: Thank you for your good suggestion. We added a short paragraph "However, the relative importance of these oxidation paths varied greatly among different researches. For instance, TMI-catalyzed oxidation could explain ~69 % of aqueous sulfate formation in NCP based on isotopic measurements and modeling (Shao et al., 2019), while oxidation by $NO_2$ or $O_2$ was the dominant oxidation path (66-73%) based on isotopic measurements in another study (He et al., 2018)" in lines 96-100 in the revised manuscript. We also added a sentence "This might be the reason why these oxidation paths showed inconsistent relative importance of among different studies even using the same method, such as isotopic measurements (Shao et al., 2019; He et al., 2018)" in lines 414-416 in the revised manuscript.

Page5 line 94: "what the control factors are from gas-phase $SO_2$ to particle-phase sulfate" does not sound good. Maybe better: "what are the controlling factors of the S(IV)-to-S(VI) conversion in the gas phase."

**Response**: Thank you. It has been corrected as you suggested "…what are the controlling factors of the S(IV)-to-S(VI) conversion from the gas phase to the particle phase" in lines 105-106 in the revised manuscript.

Page5 line 95-97: These two sentences here do not fit, or a transition sentence is missing.

**Response**: Thank you. We added a new sentence between these two sentences "This will modify its physical properties, such as morphology, phase-state and so on" in lines 108-109 in the revised manuscript.

Page5 line 100: "… statistically investigated to identify the controlling factors." Subsequently, the different sensitivity investigations should be briefly mentioned. The

uptake investigations are missing in this paragraph.

**Response**: Thank you for your suggestion. We revised this sentence "…statistically investigated to identify the controlling factors. The role of mass transfer of $SO_2$ and the oxidation of S(IV) in particle-phase have been discussed based on flow tube experiments and box model simulations" in lines 113-115 in the revised manuscript.

Page5 line 101: Replace "mainly" by "strongly".

**Response**: Thank you. It has been corrected in line 117 in the revised manuscript.

Page6 line 106: Remove "Lat." and "Lon.". Instead, add "N" and "E" behind the numbers, respectively.

**Response**: Thank you. It has been corrected in lines 121 and 122 in the revised manuscript.

Page6 line 120: "Particle-phase"

**Response**: Thank you. It has been corrected in lines 135-136 in the revised manuscript.

Page6 line 120: Measured Fe and Mn concentrations are total metal concentrations or water-soluble concentrations?

**Response**: Thank you. They are total metal concentrations. This sentence has been revised "Particle-phase total concentrations of Fe and Mn were measured…" in lines 135-136 in the revised manuscript.

Page7 line 133: Define "IE".

**Response**: Thank you. It has been defined "The ionization efficiency (IE)…" in line 148 in the revised manuscript.

Page7 line 143: "influence of RH" and "the uptake coefficient ($\gamma_{SO2}$)"

**Response**: Thank you. It has been corrected in line 158 in the revised manuscript.

290    Page7 line 146: "gas-phase"

Response: Thank you. It has been corrected in line 161 in the revised manuscript.

Page8 line 153: "gas-phase"

Response: Thank you. It has been corrected in line 168 in the revised manuscript.

295

Page8 line 155: "BET" must already be defined here for the first time, and not only in line 188.

Response: Thank you so much. It has been corrected in line 170 in the revised manuscript.

300

Page9 line 183: "we aimed to understand"

Response: Thank you. It has been corrected in line 200 in the revised manuscript.

Page10 line 203-205: Why haven't the authors applied more simple approaches to

305    calculate or estimate the pH. Please see Pye et al. (2020) and proxy approaches therein. Pye, H. O. T., et al. (2020), The acidity of atmospheric particles and clouds, Atmos. Chem. Phys., 20(8), 4809-4888. https://doi.org/10.5194/acp-20-4809-2020.

Response: Thank you. Actually, the calculation method for pH in this work is the same as that used in the literature (Pye et al., 2020). Because the unit of $H^+$ is $\mu g\ m^{-3}$ in the

310    output file of ISORROPIA II model, we need to convert it to molality. We revised

equation 1 "pH $= -\log_{10}(\gamma_{H^+} m_{H^+}) = -\log_{10} \dfrac{1000 \gamma_{H^+} c_{H^+}}{AWC}$    (1)

where $\gamma_{H+}$ is the activity coefficient of $H^+$ and $m_{H+}$ is the molality of $H^+$" in lines 221-222 in the revised manuscript.

315    Page10 line 207-212: Please, provide the references for all kinetic rate expressions given. Furthermore, the authors should look at two reviews published recently that have evaluated kinetic data on sulfur oxidation (Liu et al. (2021); Tilgner et al. (2021, under

review in ACPD)). Are the applied kinetic data in agreement with their recommended values?

320 Liu, T., et al. (2021), Multiphase Oxidation of Sulfur Dioxide in Aerosol Particles: Implications for Sulfate Formation in Polluted Environments, Environ. Sci. Technol., 55(8), 4227-4242. https://doi.org/10.1021/acs.est.0c06496 .

Tilgner, A., et al. (2021), Acidity and the multiphase chemistry of atmospheric aqueous particles and clouds, Atmos. Chem. Phys. Discuss., https://doi.org/10.5194/acp-2021-
325 58, in review, 2021.

**Response**: Thank you for your suggestion. The citations have been added in lines 231-232 in the revised manuscript. The kinetic data in this work are agreement with their recommended values (Liu et al., 2021;Tilgner et al., 2021). These references have also been added in line 241 in the revised manuscript.

330

Page11 line 215: "8:7" should be "8.7"

**Response**: Thank you. It has been corrected in line 239 in the revised manuscript.

Page11 line 219: Please, cite the references for all Henry's law constants.

335 **Response**: Thank you. It has been cited "$1.1 \times 10^{-2}$, $1.0 \times 10^{5}$ and $1.0 \times 10^{-2}$ M atm$^{-1}$ at 298 K for $O_3$, $H_2O_2$ and $NO_2$ (Seinfeld and Pandis, 2006), respectively" lines 243-244 in the revised manuscript.

Page11 line 219-221: Where can I find the derived $H_2O_2$ concentrations? Do they fit to
340 measurements in the NCP, see e.g. Ye et al. (2018)?

Ye, C., et al. (2018), High $H_2O_2$ concentrations observed during haze periods during the winter in Beijing: Importance of $H_2O_2$ oxidation in sulfate formation, Environ. Sci. Tech. Let., 5(12), 757-763, https://doi.org/10.1021/acs.estlett.8b00579.

**Response**: Thank you so much for your comment. Fig. R3A shows the derived $H_2O_2$
345 concentrations. Fig. R3B shows the diurnal curves of $H_2O_2$ in winter. The black line and the grey shadow are the mean values and the error bars reported by Ye et al. (Ye et

al., 2018) and the red line with error bars is the derived $H_2O_2$ concentrations in winter in this work. Overall, the $H_2O_2$ concentrations derived in this work are comparable with those reported by Ye et al.(Ye et al., 2018). In the revised manuscript, we added this figure in the SI. We also added a short paragraph "Fig. S2 shows the derived $H_2O_2$ concentrations and the diurnal curves of $H_2O_2$ in winter in Shijiazhuang. The $H_2O_2$ concentrations varied from 0.05 to 3.7 ppbv, with a mean value of 0.62±0.52 ppbv. Overall, the wintertime $H_2O_2$ concentrations derived in this work are comparable with those reported in the literature (Ye et al., 2018)" in lines 245-249 in the revised manuscript.

[Figure]

Fig. R3. (A) the derived $H_2O_2$ concentrations and (B) the diurnal variations of $H_2O_2$ in winter in Shijiazhuang.

Page11 line 235-236: Please, provide only relevant decimal places.

**Response**: Thank you. It has been corrected in line 264 in the revised manuscript.

Page12 line 238: Remove "well".

**Response**: Thank you. It has been corrected in line 266 in the revised manuscript.

Page12 line 238: "larger population of heavy industries" sounds bad. Do you mean "larger density of heavy industries"?

**Response**: Thank you. It has been replaced with "larger density of heavy industries" in line 267 in the revised manuscript.

370

Page12 line 240: "than in Beijing".

**Response**: Thank you. It has been corrected in line 268 in the revised manuscript.

Page12 Fig1: The legend is not well placed.

375 **Response**: Thank you. This problem has been fixed in Fig. 1 in the revised manuscript.

Page13 line 261: Another consequence of "the increased traffic emissions in Beijing", i.e. higher NOx emissions, is that the concentrations of ozone are elevated in Beijing. This should be mentioned!

380 **Response**: Thank you. Figure R4 shows the daily mean concentrations of $SO_2$, $NO_2$ and $O_3$ in Beijing from 2014 to 2020. The concentrations of $SO_2$ decreased obviously, while the concentrations of $NO_2$ showed a slight decrease and the concentrations of $O_3$ are relatively stable in the same season. So, we revised the sentence "This can be ascribed to the effective reduction of $SO_2$ emissions, but less effective reduction of
385 traffic emissions in Beijing" in lines 288-289 in the revised manuscript.

[Figure]

Fig. R4. Variations of daily mean concentrations of (A) $SO_2$, (B) $NO_2$ and (C) $O_3$ in Beijing from 2014 to 2020.

Page13 line 272: Better say that the Shijiazhuang site is more influenced by primary emissions.

**Response**: Thank you for your suggestion. We revised this sentence as "The high primary emissions of $SO_2$ in Shijiazhuang should lead to a lower SOR than that in Beijing" in lines 298-299 in the revised manuscript.

Page13 line 274: Please clarify "significantly higher". 55 ppb and 51 ppb are not significantly different!

**Response**: Thank you. Although the difference is not so big, it is significant with a *P* value of $9\times10^{-96}$ based on T-test. This means the difference is significant at 95% significance level.

Page14 line 275-279: I do not agree with the conclusion drawn here, because of the higher primary emissions in Shijiazhuang affecting the SOR. Perhaps other parameters are required to reach this conclusion.

**Response**: Thank you so much for your good comment. We rewrote this paragraph as "The high primary emissions of $SO_2$ in Shijiazhuang should lead to a lower SOR than that in Beijing. On the other hand, secondary transform of $SO_2$ to sulfate should also have influence on the SOR. The $O_x$ ($O_x = NO_2+O_3$) concentration in Shijiazhuang was usually higher than that in Beijing (Fig. 1F). The annual mean $O_x$ concentration in Shijiazhuang was $55.2 \pm 22.3$ ppb, which was significantly higher than that in Beijing ($50.7 \pm 21.5$ ppb) at 0.05 level. This is inconsistent with the observed higher SOR in Beijing if gas-phase oxidation mainly contributed to sulfate formation. These results suggest that heterogeneous and/or multiphase reactions may also play important roles in particulate sulfate formation during transport (Zheng et al., 2015; Martin and Good, 1991; Wu et al., 2019)" in lines 288-307 in the revised manuscript.

Page14 line 276: "gas-phase"

**Response**: Thank you. It has been corrected in line 304 in the revised manuscript.

Page14 line 276: "multiphase"

**Response**: Thank you. It has been corrected in line 305 in the revised manuscript.

Page14 line 282-283: "$PM_{2.5}$ mass concentration well kept pace with the high sulfate concentration" sounds bad.

**Response**: Thank you. We changed this sentence as "…$PM_{2.5}$ mass concentration coincided with the high sulfate concentration, the fraction of sulfate and the SOR" in lines 310-311 in the revised manuscript.

Page14 line 288: "a similar".

**Response**: Thank you. It has been corrected in line 316 in the revised manuscript.

Page15 line 288: "As shown in Fig. 2D, the high concentration of sulfate positively correlated with high RH in most cases". I'm not convinced here and it's hard to see from the Figure! Please provide a correlation coefficient.

**Response**: Thank you for your comments. Figure R5 shows the 2D Kernel density graph between the sulfate concentration and the RH. Overall, we can see a positive correlation between sulfate concentration and RH. The correlation coefficient is 0.92 between the probability weighted concentration and RH.

[Figure]

440    Fig. R5. The 2D Kernel density graph between the sulfate concentration and the RH.

Page18 line 352: "gas-phase".

**Response**: Thank you. It has been corrected in line 380 in the revised manuscript.

445    Page19 line 368: "the uptake".

**Response**: Thank you. It has been corrected in line 396 in the revised manuscript.

Page19 line 369: "a quick".

**Response**: Thank you. It has been corrected in line 397 in the revised manuscript.

450

Page19 line 373: "metals".

**Response**: Thank you. It has been corrected in line 403 in the revised manuscript.

Page19 line 381: I think Fig.S5 contains important information and should be therefore

455    part of the main manuscript.

**Response**: Thank you for your suggestion. We added it in Fig. R6A and Fig. 4A.

[Figure]

Fig. R6. (A) The relative importance of oxidation paths of S(IV) in aqueous phase, the dependence of (B) sulfate formation rates and (C) the probability weighted sulfate formation rates on RH in Shijiazhuang.

Page23 line 451: Replace "with" by "as a function of".

**Response**: Thank you. It has been corrected in line 497 in the revised manuscript.

Page23 line 457-458: Please revise the Figure caption and describe in more detail what is shown in the different items.

**Response**: Thank you. We revised the caption "Variations of (A) the mass concentrations and (B) the mass fractions of molecular composition in $PM_{2.5}$, (C) the estimated AWC attributed to different composition and (D) the corresponding AWC fraction as a function of RH in Shijiazhuang" in lines 503-505 in the revised manuscript.

Page23 line 459-461: Please see e.g. Li et al. (2017) for more recent DRH values incl. other salts. Why $(NH_4)HSO_4$ is not listed here which has a lower DRH than $(NH_4)NO_3$? Therefore, the following conclusion ("…ammonium nitrate should the major

contributor to the AWC compared with sulfate and chloride…") can be wrong and the subsequent discussion should be revised.

Li, Y. J., et al. (2016), Rebounding hygroscopic inorganic aerosol particles: Liquids, gels, and hydrates, Aerosol Sci. Technol., 51(3), 388-396. https://doi.org/10.1080/02786826.2016.1263384.

**Response**: Thank you for your good suggestion. We agree with you that $(NH_4)HSO_4$ has a lower DRH than $NH_4NO_3$. However, $NH_3$ is abundant in North China. For example, the annual mean concentration of $NH_3$ was 34.5±18.0 ppb in Shijiazhuang. Figure R2 shows the $R_{NH4+/SO42-}$ in Shijiazhuang. Only 1.6% of the data points showed the $R_{NH4+/SO42-}$ lower than 2.0. The concentrations of $NH_4HSO_4$ for 98.4% of the dataset were zero. As shown in Figure 5C, $NH_4NO_3$ and $(NH_4)_2SO_4$ are the major contributors to the AWC. Especially, $NH_4NO_3$ dominated the AWC when the RH ranged from 60 % to 80 %. In the revised manuscript, we deleted the sentence "…ammonium nitrate should the major contributor to the AWC compared with sulfate and chloride…" We also added a new short paragraph "It should be noted that $(NH_4)HSO_4$ has a lower DRH than $NH_4NO_3$ (Li et al., 2017b). However, 98.4% of the data points showed the $R_{NH4+/SO42-}$ higher than 2.0 in Shijiazhuang. This means that the contribution of $(NH_4)HSO_4$ to $PM_{2.5}$ should be negligible because of the abundance of atmospheric $NH_3$ in North China" in lines 517-521 in the revised manuscript.

Page23 line 465: Here, the E-AIM model is mentioned for the first time. Why not in Section 2? Would it be possible to use only E-AIM or ISOROPIA in the present study?

**Response**: Thank you so much. We moved the sentence "The deliquescence curves of inorganic salts were calculated at 298.5 K using the E-AIM model (Clegg et al., 1998). Then, the AWC attributed to individual salt was calculated with the mass of the salt and the mass-based growth factor at the corresponding RH" to Section 2 (lines 222-225) in the revised manuscript. ISOROPIA is a widely used model for AWC and aerosol pH calculations (Ding et al., 2019). However, the AWC attributed to different molecular component is unavailable in the outputs of the ISOROPIA model. Thus, we calculated

it using the mass-based growth factor and the mass concentration of individual salt. Thus, the E-AIM was used to calculate the growth factor. It is also a widely used model to calculate the growth factor of salts.

Page26 line 524-535: A recently submitted review by Tilgner et al. (2021, under review in ACPD) has outlined that the reaction rate constant of the $NO_2$ reaction with dissolved S(IV) by Clifton et al. (1988) is far too high and that studies by Spindler et al. (2003) showed much lower values. This fact should be also reflected in the discussion here.

Clifton, C. L., et al. (1988), Rate constant for the reaction of nitrogen dioxide with sulfur(IV) over the pH range 5.3-13, Environ. Sci. Technol., 22(5), 586-589. https://doi.org/10.1021/es00170a018.

Spindler, G., et al. (2003), Wet annular denuder measurements of nitrous acid: laboratory study of the artefact reaction of $NO_2$ with S(IV) in aqueous solution and comparison with field measurements, Atmos. Environ., 37(19), 2643-2662, https://doi.org/10.1016/S1352-2310(03)00209-7.

Tilgner, A., et al. (2021), Acidity and the multiphase chemistry of atmospheric aqueous particles and clouds, Atmos. Chem. Phys. Discuss., https://doi.org/10.5194/acp-2021-58, in review, 2021.

**Response**: Thank you so much for your good suggestion. We added the point "3) The previous calculations were conducted using a high reaction rate constant of the $NO_2$ reaction with dissolved S(IV) (Clifton et al., 1988; Cheng et al., 2016), while a small value was reported in the more recent study (Spindler et al., 2003; Tilgner et al., 2021)" in lines 581-584 in the revised manuscript.

Page28 line Fig.6: In this Figure, it would be better to use $O_3$ instead of $O_x$, because $NO_2$ is also considered separately.

**Response**: Thank you for your good suggestion. We replaced $O_x$ with $O_3$ in Fig. 6 and Fig. S10 in the revised manuscript. We also updated the corresponding text (from lines 621 to 623).

Page28 line 560: "gas-phase".

**Response**: Thank you. Thank you. It has been corrected in line 589 in the revised manuscript. We also fixed the same problems throughout the paper, such as in lines 70, 89, 161, 168, 380, 479, 620 in the revised manuscript.

Page29 line 577-579: Here, it should be mentioned that the effective solubility of $SO_2$ can be enhanced due the increase of the aerosol pH. Furthermore, a lower acidity also promotes other oxidation processes and enables therefore higher S(VI) formation rates.

**Response**: Thank you so much. We revised it as "These results further confirm that $NH_3$ can promote the uptake of $SO_2$ at high RH, possible through enhancing the solubility of $SO_2$ in water (Chen et al., 2019;Cheng et al., 2016;Wang et al., 2016) because the effective solubility of $SO_2$ can be enhanced due to the increase of the aerosol pH" in lines 637-641 in the revised manuscript. The effect of aerosol pH on oxidation rate of S(IV) was also discussed in a new paragraph "Aerosol acidity is one of important factors affecting the sulfate formation and the partitioning of semi-volatile gases in the atmosphere (Liu et al., 2021). As shown in Fig. S11, when aerosol pH is lower than 4.5, the oxidation rate of S(IV) in aerosol liquid phase decreases as a function of pH because the oxidation of S(IV) by transition metals is the dominant path and is negatively dependent on aerosol pH. However, the oxidation rate of S(IV) increases when the aerosol pH is higher than 4.5. This can be explained by the fact that the solubility and effective Henry's law constant of $SO_2$ are positively dependent on pH (Cheng et al., 2016; Liu et al., 2021; Liu et al., 2020a), which is consistent with the promotion effect of sulfate formation by $NH_3$" in lines 642-650 in the revised manuscript.

Page29 line 588: "liquid-phase".

**Response**: Thank you. It has been corrected through the paper, such as lines 659, 687 and 459 in the revised manuscript.

Page32 line 643 ff: Please check again all references. The reference style is not uniform, for example the doi style.

**Response**: Thank you so much. We fixed all the references including the doi style.

Supporting Information (SI): The Figure captions in the SI are in parts rather brief. I strongly recommend to extend the captions, especially for complex Figures with multiple items.

**Response**: Thank you for your suggestion. We extended the captions in SI. For example, the caption of Fig. S7 was revised "Correlation of the ionic charge between inorganic anions ($NO_3^-$, $SO_4^{2-}$, $Cl^-$) and cations ($Ca^{2+}$, $Mg^{2+}$, $K^+$, $Na^+$, $NH_4^+$) and (B) the relative contribution of cations to the total positive charges in soluble $PM_{2.5}$". The caption of Fig. S9 was also revised "(A) The time series of AWC calculated under different episodes and (B) the relative change of AWC due to reduction of ammonium nitrate (AN) and ammonium sulfate (AS) in $PM_{2.5}$".

**References:**

Chen, T., Chu, B., Ge, Y., Zhang, S., Ma, Q., He, H., and Li, S.-M.: Enhancement of aqueous sulfate formation by the coexistence of $NO_2/NH_3$ under high ionic strengths in aerosol water, Environ. Pollut., 252, 236-244, https://doi.org/10.1016/j.envpol.2019.05.119, 2019.

Cheng, Y., Zheng, G., Wei, C., Mu, Q., Zheng, B., Wang, Z., Gao, M., Zhang, Q., He, K., Carmichael, G., Poschl, U., and Su, H.: Reactive nitrogen chemistry in aerosol water as a source of sulfate during haze events in China, Sci. Adv., 2, https://doi.org/10.1126/sciadv.1601530, 2016.

Clegg, S. L., Brimblecombe, P., and Wexler, A. S.: Thermodynamic Model of the System $H^+-NH_4^+-Na^+-SO_4^{2-}-NO_3^--Cl^--H_2O$ at 298.15 K, J. Phys. Chem. A., 102, 2155-2171, https://doi.org/10.1021/jp973043j, 1998.

Ding, J., Zhao, P., Su, J., Dong, Q., Du, X., and Zhang, Y.: Aerosol pH and its driving factors in Beijing, Atmos. Chem. Phys., 19, 7939-7954, https://doi.org/10.5194/acp-19-7939-2019, 2019.

Liu, P., Ye, C., Xue, C., Zhang, C., Mu, Y., and Sun, X.: Formation mechanisms of atmospheric nitrate

and sulfate during the winter haze pollution periods in Beijing: gas-phase, heterogeneous and aqueous-phase chemistry, Atmos. Chem. Phys., 20, 4153-4165, https://doi.org/10.5194/acp-20-4153-2020, 2020.

Liu, T., Chan, A. W. H., and Abbatt, J. P. D.: Multiphase oxidation of sulfur dioxide in aerosol particles: Implications for sulfate formation in polluted environments, Environ. Sci. Technol., 55, 4227-4242, https://doi.org/10.1021/acs.est.0c06496, 2021.

Pye, H. O. T., Nenes, A., Alexander, B., Ault, A. P., Barth, M. C., Clegg, S. L., Collett Jr, J. L., Fahey, K. M., Hennigan, C. J., Herrmann, H., Kanakidou, M., Kelly, J. T., Ku, I. T., McNeill, V. F., Riemer, N., Schaefer, T., Shi, G., Tilgner, A., Walker, J. T., Wang, T., Weber, R., Xing, J., Zaveri, R. A., and Zuend, A.: The acidity of atmospheric particles and clouds, Atmos. Chem. Phys., 20, 4809-4888, https://doi.org/10.5194/acp-20-4809-2020, 2020.

Seinfeld, J. H., and Pandis, S. N.: Atmospheric chemistry and physics: From air pollution to climate change, Second ed., John Wiley and Sons, New Jersey, 429-44, 2006

Tilgner, A., Schaefer, T., Alexander, B., Barth, M., Collett Jr, J. L., Fahey, K. M., Nenes, A., Pye, H. O. T., Herrmann, H., and McNeill, V. F.: Acidity and the multiphase chemistry of atmospheric aqueous particles and clouds, Atmos. Chem. Phys. Discuss., 2021, 1-82, https://doi.org/10.5194/acp-2021-58, 2021.

Wang, G., Zhang, R., Gomez, M. E., Yang, L., Zamora, M. L., Hu, M., Lin, Y., Peng, J., Guoc, S., Meng, J., Li, J., Cheng, C., Hu, T., Ren, Y., Wang, Y., Gao, J., Cao, J., An, Z., Zhou, W., Li, G., Wang, J., Tian, P., Marrero-Ortiz, W., Secrest, J., Du, Z., Zheng, J., Shang, D., Zeng, L., Shao, M., Wang, W., Huang, Y., Wang, Y., Zhu, Y., Li, Y., Hu, J., Pan, B., Cai, L., Cheng, Y., Ji, Y., Zhang, F., Rosenfeld, D., Liss, P. S., Duce, R. A., Kolb, C. E., and Molina, M. J.: Persistent sulfate formation from London Fog to Chinese haze, Proc. Natl. Acad. Sci. USA, 113, 13630-13635, https://doi.org/10.1073/pnas.1616540113, 2016.

Yang, W., Ma, Q., Liu, Y., Ma, J., Chu, B., and He, H.: The effect of water on the heterogeneous reactions of $SO_2$ and $NH_3$ on the surfaces of $\alpha$-$Fe_2O_3$ and $\gamma$-$Al_2O_3$, Environ. Sci.: Nano, 6, 2749-2758, https://doi.org/10.1039/C9EN00574A, 2019.

Ye, C., Liu, P., Ma, Z., Xue, C., Zhang, C., Zhang, Y., Liu, J., Liu, C., Sun, X., and Mu, Y.: High $H_2O_2$ concentrations observed during haze periods during the winter in Beijing: Importance of $H_2O_2$ oxidation in sulfate formation, Environ. Sci. Technol. Lett., 5, 757-763, https://doi.org/10.1021/acs.estlett.8b00579, 2018.

---

## Author Response (AR2)

Dear Professor Harald Saathoff,

We appreciate your careful consideration of our manuscript. We have carefully responded to all of your point-by-point comments and have revised the manuscript accordingly. These revisions are described in detail below.

**General comments**

I think you have addressed the reviewer comments well and improved your manuscript significantly. Therefore, I have only a few final comments which I would like you to take into account.

**Response**: Thank you so much for your positive comments.

**Specific comments:**

Line 136: "…of Fe and Mn were…" -> "…of iron and manganese were…"

**Response**: Thank you. It has been corrected in line 136 in the revised manuscript.

Lines 188-189: "Around 50 % of $NH_4NO_3$ remained in the mixture due to evaporation" -> "Around 50% of the $NH_4NO_3$ remained in the mixture even after heating and potential evaporation"

**Response**: Thank you. It has been corrected in lines 188-189 in the revised manuscript.

Lines 299-300: "On the other hand, secondary transform of $SO_2$ to sulfate should also have influence on the SOR" -> "On the other hand, secondary transformation of $SO_2$ to sulfate should also have an influence on the SOR"

**Response**: Thank you. It has been corrected in lines 299-300 in the revised manuscript.

Line 400: "…4.2 calculated…" -> "…4.2 as calculated…"

**Response**: Thank you. It has been corrected in line 400 in the revised manuscript.

Lines 436-437: "It should be noted the mass transfer of $SO_2$ was not thought as the

RDS..." -> "It should be noted that the mass transfer of $SO_2$ was not assumed to be the RDS..."

**Response**: Thank you. It has been corrected in lines 436-437 in the revised manuscript.

Line 441: "…might be greatly overestimated..." -> "…might have been overestimated..."

**Response**: Thank you. It has been corrected in lines 441-442 in the revised manuscript.

Lines 510-511: "…salts overall underestimated around 13 % of that calculated..." -> "…salts is underestimated by around 13% compared to that calculated..."

**Response**: Thank you. It has been corrected in lines 510-511 in the revised manuscript.

Line 583: "…while a small value was..." -> "…while a smaller value was..."

**Response**: Thank you. It has been corrected in line 583 in the revised manuscript.

Line 642: "Aerosol acidity is one of important factors..." -> "Aerosol acidity is one of the important factors...."

**Response**: Thank you. It has been corrected in line 643 in the revised manuscript.

Lines 644-646: "…aerosol liquid phase decreases as a function of pH because the oxidation of S(IV) by transition metals is the dominant path and is negatively dependent on aerosol pH" -> "……aerosol liquid phase decreases with decreasing pH because the oxidation of S(IV) by transition metals is the dominant path and is decreasing with aerosol pH"

**Response**: Thank you. It has been corrected in lines 645-647 in the revised manuscript.